# network-based constraint to evaluate climate sensitivity

Lucile Ricard[1], Fabrizio Falasca[2], Jakob Runge[3,4,5] & Athanasios Nenes ●[1,6] ✉

The 2015 Paris agreement was established to limit Greenhouse gas (GHG) global warming below 1.5°C above preindustrial era values. Knowledge of climate sensitivity to GHG levels is central for formulating effective climate policies, yet its exact value is shroud in uncertainty. Climate sensitivity is quantitatively expressed in terms of Equilibrium Climate Sensitivity (ECS) and Transient Climate Response (TCR), estimating global temperature responses after an abrupt or transient doubling of $CO_2$. Here, we represent the complex and highly-dimensional behavior of modelled climate via low-dimensional emergent networks to evaluate Climate Sensitivity (netCS), by first reconstructing meaningful components describing regional subprocesses, and secondly inferring the causal links between these to construct causal networks. We apply this methodology to Sea Surface Temperature (SST) simulations and investigate two different metrics in order to derive weighted estimates that yield likely ranges of ECS (2.35–4.81°C) and TCR (1.53-2.60°C). These ranges are narrower than the unconstrained distributions and consistent with the ranges of the IPCC AR6 estimates. More importantly, netCS demonstrates that SST patterns (at "fast" timescales) are linked to climate sensitivity; SST patterns over the historical period exclude median sensitivity but not low-sensitivity (ECS < 3.0°C) or very high sensitivity (ECS ≥ 4.5°C) models.

"Climate sensitivity", or the change in the surface temperature of the Earth under increased emissions of $CO_2$, is a crucial quantity for climate policy decisions. Two metrics are most often used to express it: the Equilibrium Climate Sensitivity (ECS), which is the temperature response after an abrupt doubled amount in $CO_2$, and the Transient Climate Response (TCR), which is the response after a transient 1% annual increase in $CO_2$ over 70 years. Although the assessed ranges of ECS and TCR are narrower in the latest IPCC report (based on CMIP6 ensemble[1]) compared to the previous one (based on CMIP5 ensemble[2]) thanks to an unprecedented combination of lines of evidence[3], there is no systematic convergence in model estimates of climate sensitivity[4]. ECS distribution derived from CMIP6 ensemble is larger compared to

the one derived from CMIP5 ensemble, with the upper bound of ECS distribution shifting towards higher values[5]. High values of climate sensitivity imply a much stronger reduction in $CO_2$ emissions required to avoid drastic and accelerating climate change; the large range of sensitivities by models imply it is highly challenging to develop effective and sustainable policies without further constraining which simulations (hence estimates of ECS, TCR) are more likely.

The range of uncertainty in ECS and TCR is currently thought to be reduced through the usage of an Emergent Constraint (EC), which consists of an explicit and statistically significant linear relationship between a constrained observable $X$ which may be either a trend or a variation in the observational period[6] (e.g. temperature variability

[1]Laboratory of Atmospheric Processes and their Impacts (LAPI), Ecole Polytechnique Fédérale de Lausanne (EPFL), Lausanne, Switzerland. [2]Courant Institute of Mathematical Sciences, New York University, New York, NY, USA. [3]German Aerospace Center, Institute of Data Science, 07745 Jena, Germany. [4]Technische Universität Berlin, Berlin, Germany. [5]Center for Scalable Data Analytics and Artificial Intelligence (ScaDS.AI) Dresden/Leipzig, TU Dresden, Dresden, Germany. [6]Center for the Study of Air Quality and Climate Change (CSTACC), Institute of Chemical Engineering Sciences, Foundation for Research and Technology Hellas (FORTH), Patras, Greece. ✉e-mail: athanasios.nenes@epfl.ch

metric[7], lower tropospheric mixing index[8], southern Intertropical Convergence Zone index[9], difference in total cloud fraction between tropics and southern midlatitudes[10]) and a variable $Y$ that relates to future climate (e.g., ECS and TCR[6]). The statistical relationship is combined with observations to constrain the probability distribution of ECS and TCR. As it relies on one metric, an EC may produce strong yet overconfident constraints, and application of different EC may give quite different results[11]. To address this issue, diverse "lines of evidence", including constraints from the instrumental records, climatology, models and constraints based on paleoclimate[3] are used to estimate climate sensitivity. These lines of evidence are combined in the assessed ranges of ECS and TCR provided in the IPCC AR6[12]. Recent studies developed an EC based on historical simulations from CMIP6 models, notably on the past global temperature trend and variability[7,13,14]. These studies conclude that the Earth's surface will warm less than currently expected in ref. 12, because their 17th−83rd percentile ranges of ECS/TCR are narrower and lower compared to their corresponding unconstrained distribution, but also compared to the latest IPCC likely ranges. SST is one of the most important of climatology variables that models need to capture to reproduce the observations for the 20th century warming[15]. While the confidence in the historical trajectory of global mean temperatures is high in models, the confidence in regional scale SST variability is lower and there are large discrepancies between the observed and modeled SST trends in warming patterns, with internal variability failing to account for these biases[16].

Here, we seek to evaluate climate sensitivity with the network of SST patterns that emerges from model simulations. The emergent networks and resulting constraint (here on called netCS) represent SST modes of variability and teleconnections (Fig. 1). Climate networks are here formulated in terms of spatial patterns (nodes) and connectivity patterns (links) that emerge from decades of data. The utilization of emergent networks to evaluate climate sensitivity is supported by the dependence of the climate sensitivity estimates to the evolving patterns of surface warming[17]. In the past years the climate sensitivity estimates derived from climate models were substantially higher from the ones derived from observed warming record and radiative balance[18]. This discrepancy arises from the large inter-model spread in radiative feedback, itself dominated by the spread in the magnitude of the pattern effect on slow time scales[19]. Here, we ensure the link with ECS and TCR by characterizing the deviation of SST anomalies to reference ones in patterns. In other words, the emergent networks and

their statistics can be used to evaluate the climate system feedbacks in fully coupled CMIP models[3]. Finally, the Fluctuation Dissipation Theorem (FDT) provides the basis to support the link between the network properties and climate sensitivity[6,20]. According to the FDT, the natural fluctuations observed in a climate variable can explain the mean response of that variable to external forcing. The variability of a system may be linked to its sensitivity[6]. Here, we state that the behavior of the network−build upon SST variability - is a good proxy of the behavior of the model under an increase of $CO_2$. Additionally, we state that the use of a causal framework[21] to infer causal links makes the EC more robust, since the causal predictors are more likely to hold under future climate change scenarios[22]. The netCS constraint is based on quantitatively comparing the simulated and observed dynamics of SST patterns. Unlike ECs, the network-based constraint relies on multiple metrics that characterize the network. Hence, netCS does not converge necessarily towards one group of CMIP6 models (i.e. low, intermediate or high climate sensitivity group), but can discriminate models with less plausible SST and connectivity patterns, and to the extent that the latter affect ECS/TCR, help constrain climate sensitivity. The challenge in such an approach relies in the ability to extract knowledge from huge amounts of large and interdependent datasets. Knowledge Discovery and Data Mining (KDD) algorithms are well placed for this purpose as they can ingest large temporal datasets and yield simplified, low-order representations of the simulations that maintain their key dynamical aspects. netCS relies on the KDD method δ-MAPS[23–25] to identify coherent regions of SST, that constitute nodes of networks. These networks are low-order graph representation of CMIP6 SST fields, and captures the underlying dynamical system. As a second step, causal links are identified with causal discovery algorithm PCMCI, which is an adaptation of the condition-selection PC algorithm (named after its inventors Peters Spirtes and Clark Glymour) followed by the Momentary Conditional Independence (MCI) test[22]. A causal model evaluation has already been successfully applied in ref. 26, in which the use of causal discovery algorithm gives more in-depth interpretation of the interaction between components of the system. As a third step, netCS estimates distances between networks of simulations and the reference networks (derived from reanalysis) to rank models. Weights are derived from metrics to emphasize the most skillful and realistic models[27–29]. netCS finally provides weighted estimates of ECS/TCR which are narrower and lower than the unweighted ones. Overall, our work addresses two important issues: the application of recent data-driven methods on an unprecedented amount of historical

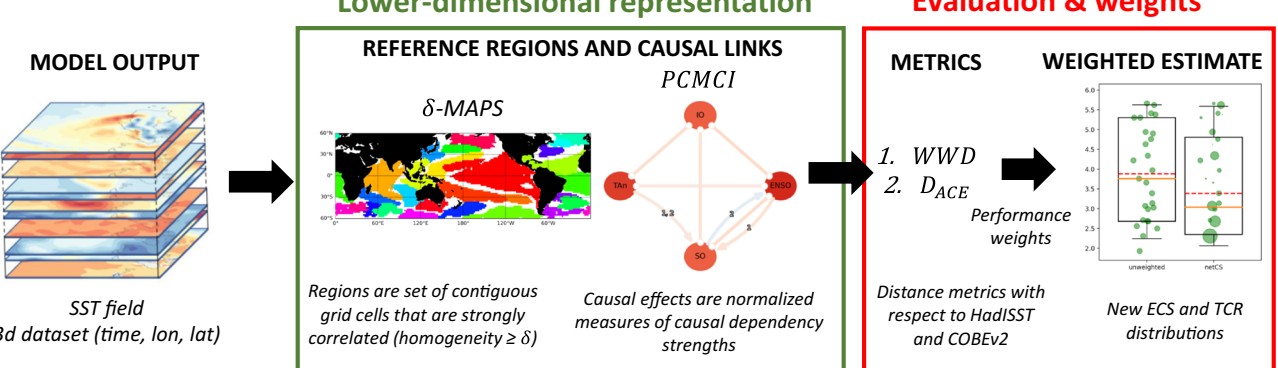

**Fig. 1 | Schematic of the network-based constraint to evaluate Climate Sensitivity (netCS) approach.** First step is the mapping of gridded data into a lower-dimensional representation using Knowledge Discovery and Data mining algorithms. From large and complex Sea Surface Temperature (SST) simulations, it infers SST regions with δ-MAPS and discovers causal structure with the causal discovery algorithm PCMCI. The Tigramite approach allows to estimate the causal effects. Second step is the evaluation of the network with respect to reference networks (reconstructed from HadISST and COBEv2). Two distance metrics are proposed. Weighted Wasserstein Distance ($WWD$) evaluates the patterns and the Distance Average Causal Effect ($D_{ACE}$) evaluates the connectivity patterns between the main regions of the climate system. A weighting scheme approach[27–29] converts distances into weights, which allows to constrain both Equilibrium Climate Sensitivity (ECS) and Transient Climate Response (TCR) with new weighting distributions.

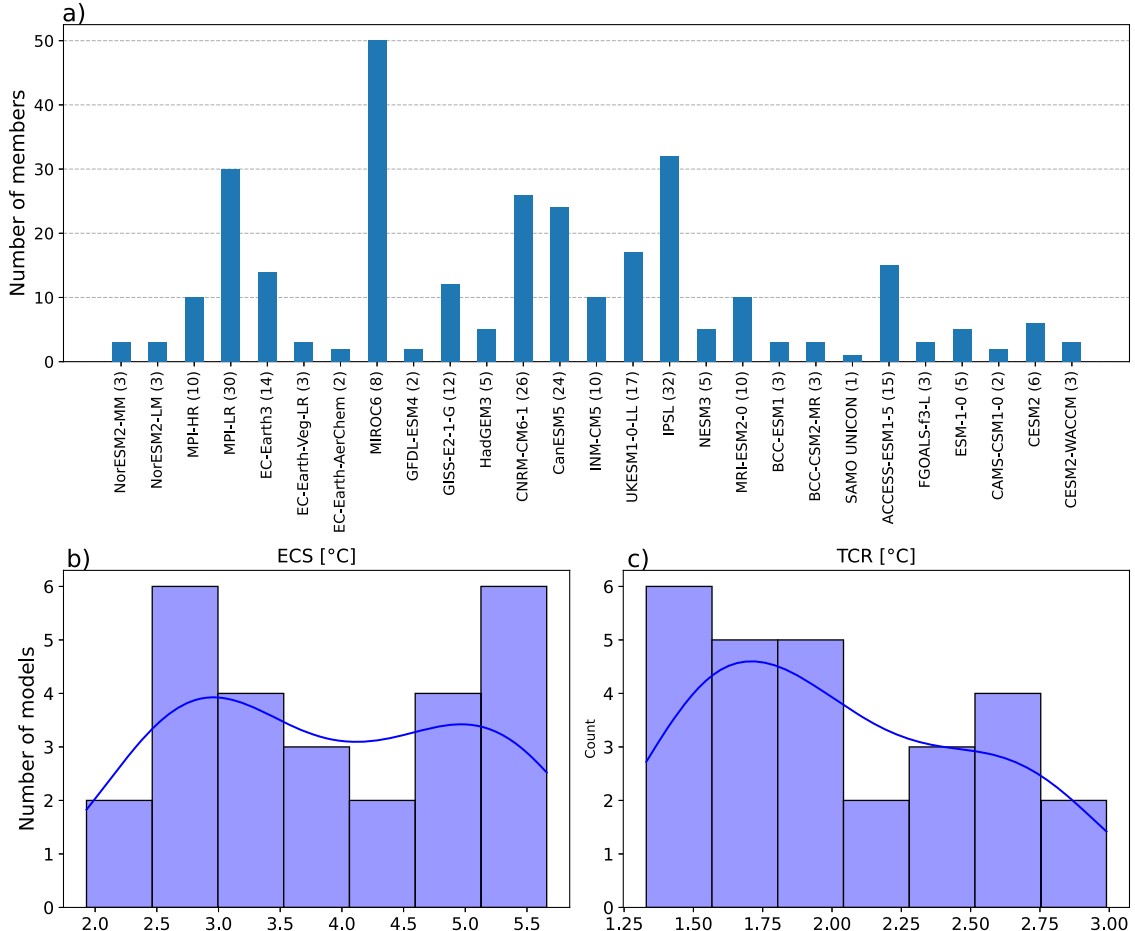

Fig. 2 | **Sea Surface Temperature and Climate Sensitivity data. a** Number of historical simulations per model, for each CMIP6 model. On average there is 11.2 members per model. Model MIROC6 is the largest ensemble with 50 members, and SAMO-UNICON is the only model contributing with one realization. From the ensemble of datasets (301 in total), we aim to extract properties to constrain the target metrics Equilibrium Climate Sensitivity (ECS) and Transient Climate Response (TCR). Density distributions of ECS (**b**) and TCR (**c**) reconstructed from the ensemble of CMIP6 models. Both distributions are bimodal with medians respectively equal to 3.76 °C and 1.92 °C.

simulations, with the ability to deal with their high-dimensionality, the time delays and the strong autocorrelations inherent to the timeseries[22], and the analyze of SST data in spatial patterns and connectivity patterns as a proxy to SST response under global warming. This type of constraint derived from an emergent network should be added as a new line of evidence of the climate sensitivity estimates.

## Results

### Dimension reduction with δ-MAPS

We analyze an ensemble of 27 CMIP6 models (Fig. 2a and Supplementary Table 1) and their ensemble of monthly historical simulations of Sea Surface Temperature. ECS and TCR values of the models are known[13]. Although the normality assumption is not rejected for ECS/TCR distribution (Fig. 2b, c and Supplementary Note 1), we can identify three groups of models: eight low-sensitivity models (ECS < 3.0 °C), nine medium-sensitivity models (3.0 ≤ ECS < 4.5 °C) and ten high-sensitivity models (ECS ≥ 4.5 °C). We constrain the distributions of climate sensitivity with properties of emergent networks, reconstructed from SST anomalies from global simulations. The analysis is limited to the tropics and subtropics (60°S – 60°N). Also, we focus on a past period spanning from 1975 to 2014, for which the uncertainty from aerosol climate forcing is largely reduced[14].

The mapping of SST data into networks constitutes the first part of netCS, whose complete flowchart is described in Fig. 1. A first algorithm, δ-MAPS, is applied to reduce the dimensionality of the spatiotemporal datasets[25]. Grid cell temperatures are not independent from their neighbor values; therefore, it makes sense to merge grid cells that covary over time into regions that have a specific role in the time-evolving climate system. δ-MAPS identifies such SST regions, as homogenous regions of high variability in the dataset, so that the map of regions (Fig. 3a) is an indicator of the consistency in the spatial grid. We consider the HadISST[30] dataset as reference at resolution of 2 by 2 degrees. The dimensionality of the dataset is reduced by δ-MAPS from 10800 grid cells to 30 regions. We can associate each SST region to its SST signal, which is defined as the area-weighted cumulative SST anomalies of the grid cells of the regions (further discussed in Methods section). Grid cells are weighted here with the cosine of their latitude. The focus is on SST time series at regional scales (Fig. 3c). Beyond region inference, links may also be inferred between regions with δ-MAPS using a time-lagged correlation analysis[25], in order to represent the underlying dynamical system of SST outputs into a directed and weighted network. The notion of strength (S) is also proposed in ref. 25 to summarize the importance of a given region within the network. S is calculated as the sum of the covariances a given region shares with all the other regions in the network. Regions have various shapes and sizes. Strength of regions gives us a good insight on which regions are the climatically-relevant regions among the regions identified[31], and only few regions out of the thirty regions contribute to the total Weighted Wasserstein Distance (*WWD*) value (Supplementary Fig. 1). Largest and strongest regions are located between 30°S and 30°N

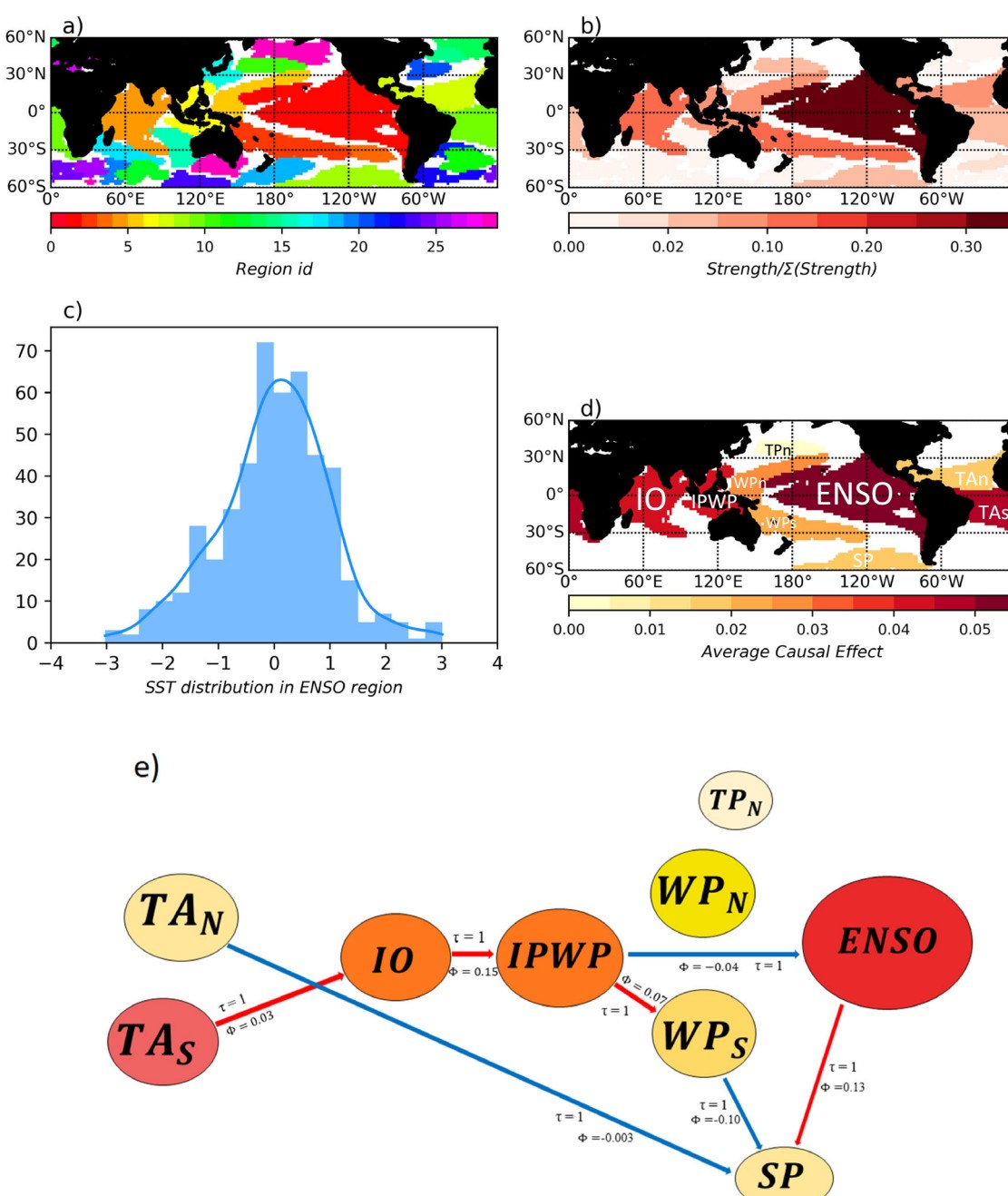

**Fig. 3 | Regions and links inference with δ-MAPS and PCMCI. a** Region map extracted from HadISST dataset with δ-MAPS algorithm. Thirty regions are identified and associated to a signal (cumulative Sea Surface Temperature anomalies). **b** Corresponding weight map: each region is colored as function of its weight (sum of covariances with all other regions divided by total sum of covariance). The region with biggest weight lies in the central east Pacific Ocean, and is referred to as the El Nino Southern Oscillation(*ENSO* region. **c** SST anomalies in ENSO region in HadISST. **d** Nine regions which serve as nodes for the causal network, labeled with their names: the *ENSO* region, the Indian Ocean (*IO*) region, the Indo-Pacific Warm Pool (*IPWP*) region, the region in South Pacific (*SP*) Ocean, the regions in the Western Tropical Pacific Ocean in the north and south hemisphere (*WP_n* and *WP_s*),

the regions in the Tropical Atlantic Ocean in the north and south hemisphere ($TA_n$ and $TA_s$) and the region in north of the Tropical Pacific ($TP_n$). Regions are colored as function of their Average Causal Effect (*ACE*) value in the reference HadISST dataset. The *ENSO* region has the largest *ACE* (**e**) Causal network reconstructed from nine regions with PCMCI algorithm, and its reference causal effects inferred in HadISST dataset with a lag of one month. Only causal paths at lag one month are displayed here. Link are labeled with the value of the lag $\tau$ in months and the value of the path coefficient $\Phi$ which is equal to causal effect for direct causal path. Red and blue link respectively indicate positive and negative causal effects (sign of the path coefficient $\Phi$). Nodes are colored as function of the *ACE* displayed on (**d**).

(Fig. 3b), and consist of convectively-active areas which in turn contribute to the variance in the climate sensitivity estimates derived from models[8]. The strongest and largest region which lies in the central east Pacific Ocean corresponds to the well-known El Niño-Southern

Oscillation (ENSO) mode. There are also regions corresponding to the Indo-Pacific Warm Pool, the western Pacific Ocean and eastern Indian Ocean which were notably qualified as warming patterns in which important discrepancies between observed SST trends and modeled

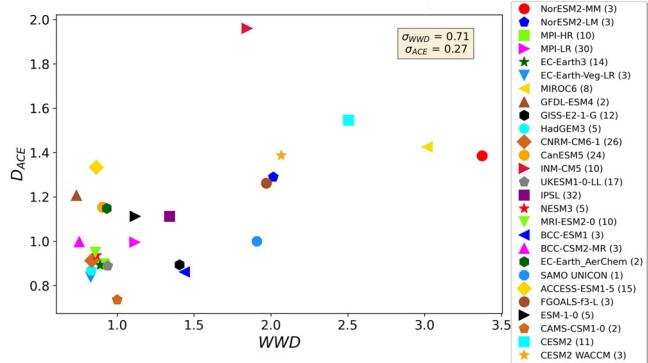

**Fig. 4 | 2D metric space <$WWD$, $D_{ACE}$>.** Each of the 27 CMIP6 model is represented by a marker with its ensemble mean Weighted Wasserstein Distance ($WWD$) value on the x-axis and its ensemble mean Distance Average Causal Effect ($D_{ACE}$) value on the y-axis. $WWD$ and $D_{ACE}$ values are the mean values of the metrics computed with respect to both HadISST and COBEv2. Number in parenthesis in the legend refers to the number of ensemble members. The ground truth is represented at the bottom left corner (0,0): the closer the marker is to the origin, the more realistic the model. To evaluate the performance of a new model, one can compute its $WWD$ and $D_{ACE}$ values and locate the model in the 2D metric space as a first assessment. $WWD$ and $D_{ACE}$ are moderately correlated, with Pearson correlation coefficient equals to 0.62.

SST trend[16] were identified. Overall, the spread of regional temperature anomalies across the models in the strongest regions identified could drive the diversity of ECS/TCR estimates across the models[18,19].

## Causal inference with PCMCI

Next step is the estimation of causal links and associated causal effects between regions (Fig. 1). Links between geographically separated regions are also referred to as "teleconnections", where fluctuations in one region can influence climate conditions in distant regions. The PCMCI algorithm[32] is applied to the monthly time series of nine regions in HadISST dataset (Supplementary Fig. 2) which appeared among the strongest regions in Fig. 3b. We chose to uncover the causal structure from only a subset of regions in order to discriminate the models based on the main teleconnections of the climate system, directly between regions that are climatically-relevant (Fig. 3d, e). PCMCI performs conditional independence tests between the time series in order to remove spurious associations that can be explained by other nodes in the graph, either as confounders, or as mediators of indirect associations. A causal interpretation of the discovered network rests on three key assumptions: one, stationary time series, second, that the underlying causal relations leave an imprint in the conditional independence structure between the nodes, and third, that there are no unobserved confounders. While the first assumption is fulfilled due to our data preprocessing (see Methods) and second assumption is a reasonable approximation, the latter cannot be assured here and, hence, the reconstructed network represents rather an approximation, where many spurious links are removed. While time-lagged links are oriented forward in time, with a maximum time lag $\tau_{\max}$ equals to three months, we leave contemporaneous links (at lag zero) unoriented here. Atmospheric teleconnections occur over a wide range of time scales through wave propagation, but we can capture the fastest over a range of three months[33].

We focus on the SST perturbations that propagate at "fast" time scales (up to 3 months), and allow the resulting connectivity between regions to bring out teleconnections with "slower" timescales. Our goal here is not to perfectly estimate the causal relations, but rather to obtain a better approximation than networks based purely on correlations as a metric to compare models with observations. We consider linear relationships between the variables, which renders the conditional independence test, here Partial Correlation test, suitable for our

analysis. Out of the total potential number of cross-links, 16% of cross-links only were revealed with a significance threshold α sets to 0.05 (Supplementary Table 2). Figure 3e demonstrates the causal network reconstructed from the nine regions with a one-month time lag. The magnitude of the links (or path coefficient Φ) is inferred with a normalized measure of the strength of the causal dependencies. This measure relies on the MCI partial correlation test statistic (for an interpretation, see ref. 32). For example, Fig. 3e shows that a perturbation of one standard deviation of the *ENSO* signal would lead to a positive perturbation of the signal in the *SP* region, with a magnitude of 0.13 (in units of its standard deviation).

## CMIP6 model performance over the historical period

To quantify how realistic CMIP6 models are compared to the reference, we compare the networks reconstructed from all model outputs (Fig. 2a and Supplementary Table 1) to the networks reconstructed from the HadISST and COBEv2 reanalysis datasets (used as reference networks as in ref. 24). For this step, we consider solely the regions identified in HadISST. This allows to fix the nodes of the network and compare only the SST signals among them. Our aim is to provide a global evaluation of the models based on their regional performances. To do so, we introduce two global-level distance metrics built upon two regional-level measures (Supplementary Fig. 3). The distance metrics to quantify the distance from the networks of the simulations to the reference networks are the *Weighted Wasserstein Distance* ($WWD$) and *Distance Average Causal Effect* ($D_{ACE}$). $WWD$ is the weighted mean of the Wasserstein Distance ($WD$) values measured in the regions, and evaluates the distance between in-region SST distributions of the model output and of the reference $D_{ACE}$ is based on the error of the Average Causal Effect ($error_{ACE}$) values measured in regions, and scores the causal effects along links of the network with respect to the true causal effects (Fig. 3e). Testing for differences in causal effects allows to capture differences in how perturbations propagates along the climate network (both metrics are further discussed in Methods section). $WWD$ and $D_{ACE}$ values are calculated for each historical simulation with respect to the two reference datasets. The agreement between the value of each metric with respect to the HadISST and COBEv2 datasets is very high (correlation coefficient equal to 1.00 and 0.97 for $WWD$ and $D_{ACE}$, respectively). This means that models rank similarly with respect to both reference datasets, and so average metrics (with respect to both reference datasets) are used. The intra-and inter-model variability of both metrics is shown in Supplementary Fig. 4, for which the spread between models is larger than the spread within models with the exception of five and eight models for $WWD$ and $D_{ACE}$. It is important to combine metrics that express different structural aspects of the networks, to provide a robust evaluation of the models. The 2D metric space reconstructed from the 27 ensemble means (Fig. 5) evidences that a model may capture well the SST distributions inside each region but not necessarily their connectivity patterns, and vice versa. The correlation between the distance metrics is equal to 0.62, which is a good agreement. Notably, there is a cluster of models (next to the origin on Fig. 4) which has both realistic SST patterns and connectivity patterns. The performance of the models over the historical period is an important step towards constraining the ECS and TCR, as models that reproduce the past period could contain better physics (or are better tuned) although not guaranteed to be the best model for thefuture[34,35], provides an important line of evidence especially if evaluation is done based on climate-relevant parameters that are less subject to tuning, such as detrended patterns of SST. This underlying premise had already been formulated[22], saying that today's dependencies between the predictors and climate sensitivity represent actual physical processes that also hold under future climate change. The ability of models to reproduce the SST patterns in the recent past period is therefore a good proxy of their ability to represent the (at least near) future, and

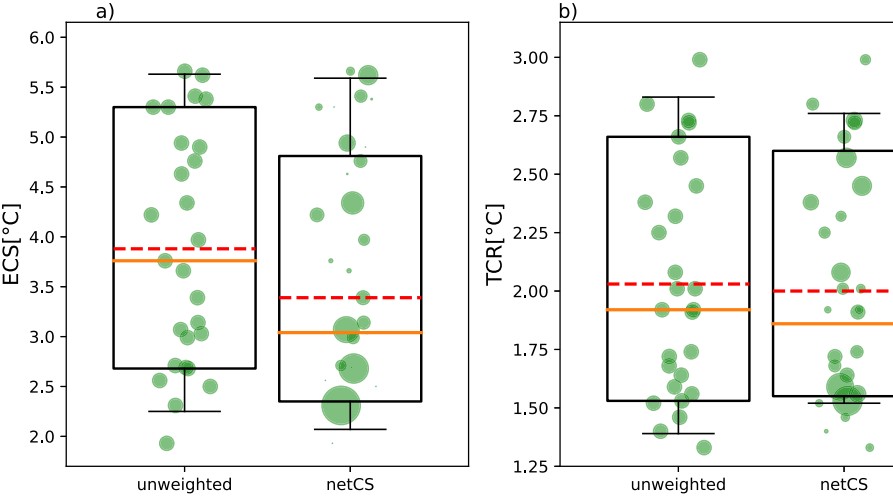

**Fig. 5 | netCS estimates of Equilibrium Climate Sensitivity and Transient Climate Response.** Central estimates and likely ranges for Equilibrium Climate Sensitivity (ECS) (**a**) and best estimates and likely ranges for Transient Climate Responses (TCR) (**b**). Unweighted and weighted ECS estimates and TCR estimates obtained with netCS approach. ECS and TCR values are represented along the y-axis. Green dots represent the ensemble mean values of the ECS/TCR of the 27 CMIP6 models and their sizes are proportional to their weights. Weighted distributions 'netCS' are derived from Weighted Wasserstein Distance ($WWD$) and Distance Average Causal Effect ($D_{ACE}$) metrics. Means (dashed red lines), medians ('central estimate'—orange lines), percentiles 17-83th ('likely range'—boxes) and

percentiles 5–95th ('very likely range'—whiskers) are represented. Central estimates are equal to 3.04 °C for ECS and to 1.86 °C for TCR. Central estimates and likely ranges are lower and narrower than the unweighted ones for both ECS and TCR. CMIP6 models represented are NorESM2-MM, NorESM2-LM, MPI-ESM1-2-HR, MPI-ESM1-2-LR, EC-Earth3, EC-Earth3-AerChem, EC-Earth3-Veg-LR, MIROC6, GFDL-ESM4, GISS-E2-1-G, HadGEM3-GC31-LL, CNRM-CM6-1, CanESM5, INM-CM5, UKESM1-0-LL, IPSL-CM6A-LR, NESM3, MRI-ESM2-0, BCC-ESM1, BCC-CSM2-MR, SAM0-UNICON, ACCESS-ESM1-5, E3SM-1-0, FGOALS-f3-L, CAMS-CSM1-0, CESM2, CESM2-WACCM.

---

we can use the different scores to downweigh underperforming CMIP6 models.

## A holistic constraint on ECS and TCR

Distance metrics are converted into performance weights using the non-linear transformation (Eq. (1)) proposed in the weighting scheme approach Climate model Weighting by Independence and Performance (ClimWIP)[27–29]. Weights are spread over a gaussian function, so that models closer to the reference get larger weights. The width of the gaussian is calibrated with the shape parameter $\sigma_D$, that depends on the observable $WWD/D_{ACE}$ and the climate sensitivity metric ECS/TCR, and is determined with a calibration approach also known as perfect model test[28] or leave-one-out cross-validation (Supplementary Note 2). The shape parameter is selected within the [20%, 200%] range of the median of the distance metric. For each $\sigma_D$ value, we assume one by one that a model's climate sensitivity is the true value and apply a weighting scheme to the remaining models. The $\sigma_D$ value is retained if 70% of the assumed "true" models fall withing the 10−90 percentile range of the weighted distribution of climate sensitivity. The final $\sigma_D$ value is the minimal value of the possible $\sigma_D$ values, which provides the most aggressive weighting. We note that the shape parameter is not adjusted to optimize the relationship between the observable and climate sensitivity. Instead, it is tuned to maintain a specific consistency when models are excluded. This calibration approach ensures that we do not instill overconfidence in models, which is a pitfall mentioned in ref. [6]. In our study, $\sigma_D$ values range between 0.33 and 1.07 (Supplementary Table 3).

$$w_i = e^{\frac{-D_i^2}{\sigma_D^2}} \qquad (1)$$

Finally, weights derived from $WWD$ and from $D_{ACE}$ are averaged and normalized (Supplementary Figs. 5 and 6). Resulting weights rely on two different structural aspects of CMIP6 models (Supplementary Fig. 7). Their variability is shown on Supplementary Table 4. The weighted distributions for both ECS and TCR are represented with

boxplots next to the boxplots of the unweighted distributions (Fig. 5). Boxplots show the mean, median, likely range (17–83 percentile) and very likely range (5-95 percentile), and the most realistic models are indicated by the larger weight (biggest size of circle). Boxplots with respect to only HadISST or only COBEv2 are shown on Supplementary Fig. 8. New likely range of ECS is shifted towards lower values compared to the unweighted range, while the likely range of TCR is squeezed (Fig. 5). More specifically, 17th percentile, median and 83th percentile of weighted ECS are lowered down compared to the ones of unweighted ECS. And median and 83th percentile of weighted TCR are lowered down when compared to the unweighted TCR, while the lower bound 17th percentile slightly increased. Climate sensitivity is constrained, because the spread of values has been reduced. Two clusters of models reproduce best the SST variability—the low-sensitivity models and the high-sensitivity models -, and the dominant one is persistently in the low ECS/TCR, which may indicate that the warming will be less that the average one from the models (Supplementary Fig. 9). This result goes in the same direction than the ensemble of studies on climate sensitivity that use the historical simulations in multi CMIP6 models[7,13,14]. It is worth noting our estimates can rigorously be compared only to the equivalent unweighted distribution, which can differ depending on the CMIP6 models each author work with. Our unweighted distribution is more spread compared to unweighted distributions of other studies, which impedes the meaningful comparison of the final estimates. This is why our likely ranges are higher than ones in previous studies (Supplementary Fig. 10). The comparison with IPCC estimates is even more difficult because the ECS and TCR ranges of the 2021's IPCC report are assessed with different techniques (paleoclimate records, climatology, observed warming in response to forcing, inferred from GCMs)[27], but the likely ranges remain consistent (Supplementary Fig. 10 and Supplementary Tables 5 and 6).

Additionally, the repartition of the models with largest weights (more realistic models) is well balanced between group of "warm models" (ECS ≥ 4.5 °C) and group of low-sensitivity models (ECS < 3 °C) (Fig. 5 and Supplementary Fig. 9). The low-sensitivity group is larger in

number and better reproduces the simulations, which is why the median and range of ECS and TCR is on the low side, but the high sensitivity models cannot be excluded. The most downweighed models are the ones with medium ECS (between 3 and 4.5 °C).

## Discussion

We proposed to use causal networks as characteristics of historical climate that are relevant to rank the models and give more reliable future projections. We estimated networks from simulated fields of SST to capture the dynamics of the climate model outputs, by inferring the main components and the systematic teleconnection patterns between them. The models are performing differently depending on the metrics we use, and we finally propose robust estimates of ECS and TCR derived from a combination of two metrics, converted into weights. netCS approach assesses ECS to be comprised between 2.35 and 4.81 °C with central estimate 3.04 °C, and TCR between 1.55 and 2.60 °C (with central estimate 1.86 °C). The consistency between the estimates with respect to HadISST and to COBEv2, and other sensitivity tests are discussed in the Supplementary Information. For future evaluation of a new model or a new parameterization, the 2D metric space $<WWD, D_{ACE}>$ can directly be used to evaluate the distance between a given model to reference (Fig. 4). Overall, the ability of models to simulate realistic SST distributions at regional scales should be incorporated in the model evaluation. Both weighted estimates of ECS and TCR are lower and narrower compared to their unweighted distributions. Despite getting constrained distributions, no group of models could be definitely excluded. Future studies could examine more closely the characteristics of each clusters' simulations and point to any systematic differences between clusters as a potential driver of ECS and TCR differences. Such clusters were not revealed in previous studies using traditional EC framework, in which procedures does not aim to discard any models. Our recommendation would be to combine netCS with another EC in order to determine which group of models is the most plausible one.

We have shown the development of netCS based on KDD algorithms, as an alternative to traditional EC techniques which tend to assume a linear relationship between the observable and the climate sensitivity metric ECS/TCR. Here we showed we can efficiently reduce the dimensions of hundreds of datasets to extract the main components and their directed effects along their long-distance interactions. We notably tried to fill the gap between our physical understanding of the Earth System (causal-based) and our common statistical tools (correlation-based), by applying to real-world data the causal discovery algorithm PCMCI. From a practical point of view, the δ-MAPS and PCMCI methods are fast to execute and give us access to interpretable metrics. Potential of network analysis and causality have not been yet fully exploited, and probably there is still lots of things to learn with such methods

The network-based constraint provides a new line of evidence for constraining climate sensitivity, as it provides a way to use high-dimensional climate data (SST patterns and connectivity specifically) to help determine those models for which ECS and TCR estimates are most plausible. An important aspect of netCS lies on whether SST connectivity patterns inferred at fast timescales (from historical period simulations) carry enough discriminatory power for constraining ECS/TCR. Based on the argument that detrended SST pattern errors reflect important errors in surface energy and water vapor fluxes that strongly affect clouds and their radiative feedbacks[36] (which are key components of climate sensitivity), we show that this is the case. More work can focus on expanding netCS to include slower interactions, other climate-relevant property networks, multiple time periods, climate regimes or focus on different aspects and metrics of global change. We therefore encourage the climate and data science community to build upon our findings and tools to further develop network-based constraints for climate science applications.

## Methods

### Climate sensitivity data

Both ECS and TCR values were obtained from Nijesse[13], in which equilibrium temperatures were extrapolated from the warming 150 years after the quadrupling of $CO_2$. ECS values spread from 1.93 °C to 5.66 °C around a median equal to 3.76 °C. TCR values are smaller and non-linearly related to ECS values because of the non-linear cooling effect of the ocean heat uptake. TCR values range from 1.33 °C to 2.99 °C around a median equal to 1.92 °C.

### SST datasets

We use the Hadley Centre Sea Ice and Sea Surface Temperature dataset (HadISST) and the COBE-SST2 dataset (COBEv2) as ground truth. HadISST dataset has been developed at the Met Office Hadley Centre for Climate Prediction and Research[30] and incorporates in situ and satellite data. The product COBEv2 is provided by the Physical Sciences Laboratory of the National Oceanic and Atmospheric Administration[37] (NOAA/OAR/ESRL, PSL, Boulder, Colorado, USA, at https://psl.noaa.gov/data/gridded/data.cobe2.html).

Climate model outputs from CMIP6 models (Supplementary Table 1) were retrieved on the Earth System Grid Federation (ESGF) nodes. We base our study on historical simulations of global mean SST, as it governs the ocean-atmosphere interactions and is at the base of many physical processes. In total we collected 301 historical simulations generated from 27 CMIP6 models, themselves provided by 21 different institutions (Fig. 2). The number of simulations provided by each model is different, that makes this ensemble an opportunity ensemble.

The data are pre-processed with Climate Data Operator[38] to obtain SST monthly anomalies over the time period 1975–2014. All simulations are bilinearly interpolated to a 2.0° × 2.0° longitude–latitude grid and in the latitude range between 60 °S and 60 °N. The final 3D datasets are made of 10 800 grid cells with a sample size equals to 480.

### Data-driven methods

The large and complex simulations of the CMIP6 models are made of thousands of points in the temporal and spatial spaces. Comparing climate change simulations is a monumental task, owing to their lengths. The methods presented here allow to drastically reduce the degree of freedom of each dataset, to get a network made of dozens of nodes and hundreds of links that represent the underlying dynamical system.

### δ-MAPS method

First, δ-MAPS algorithm identifies regions in the 3D dataset of SST anomalies. Here we briefly summarize the δ-MAPS methodology and refer to ref. 23 for an in-depth discussion. The region identification algorithm starts from a set of few *seeds*, consisting of a grid cell *i* and its k neighbors. Each grid cell embeds a time series $x_i(t)$. Seeds are then merged and expanded to form regions. The merging and expansion step allow to iteratively add grid cells to the original seeds. At each step of the algorithm, regions are defined as a set of time series with average pairwise correlation larger than a threshold δ. The two hyperparameters, the number of neighbors k and the homogeneity threshold δ, modulate the size and number of regions captured. Here we selected k equals to 8. ref. 23 proposed a heuristic for computing δ based on the average significant pairwise correlation between a sample of time series $x_i(t)$, $x_j(t)$. The significance level α adopted here equals to 0.01. The statistical test considered here addresses the challenge of testing timeseries with strong autocorrelation structure. As a second step, we associate a signal to each region. Such time series is defined as the cumulative SST anomalies, i.e. the area-weighted timeseries computed from all the timeseries of the grid cells the region is made of. For a region A, composed of *n* grid cells, its cumulative signal anomaly $x_A(t)$ is given by $x_A(t) = \sum_{i=1}^{n} x_i(t) \cos(\theta_i)$ where $\theta_i$ is the latitude of the

cell $i$. δ-MAPS has already been successfully applied to SST datasets in a multi-model assessment of CMIP6 models[24] and to 6000-year long transient simulations[39–41]. Moreover, the algorithm has a relatively fast execution time, on the order of a one or two hours with our tuning (Supplementary Fig. 11).

δ-MAPS algorithm infers the links between the regions, in order to build a network made of modes of variability and their connectivity patterns. Mathematically, links correspond to significantly lagged correlations between all pair of signals, and physically they represent the teleconnections that convey energy between two distant regions of the globe. The algorithm examines the correlations over a range of time lags between all pair of signals. The lagged correlations are computed by the Bartlett's formula to account for autocorrelations. The significant correlations are retained, and we note $\sigma^*$ the lag value that maximizes the correlation in absolute sense, and $r_{A,B}^*$ the associated correlation between two regions $A$ and $B$. $r_{A,B}^*$ indicates us the temporal ordering of the events, and allows us to give a direction to the link[25]. In the case the maximum correlation is inferred at lag 0, the link is undirected. The magnitude of the link, termed as weight, is defined as the covariance between the pair of signals and formulated as $w_{A,B} = \sqrt{\sigma_A^2 \sigma_b^2} r_{A,B}^*$. Subsequently, the strength $S$ of a region is defined as the sum of the absolute values of its weights[24]. Finally, a directed and weighted network represents the underlying dynamical system of the model outputs.

## Causal discovery algorithm PCMCI

The causal discovery algorithm PCMCI (developed in ref. 32, implemented in the Tigramite open-source software package for Python and available from https://github.com/jakobrunge/tigramite) is a data-driven approach developed to infer causal links and causal effects from large amount of data available. PCMCI estimates the causal structure from timeseries corresponding to the nodes of the system, under the following assumptions: stationarity, which can reasonably be assumed for the SST monthly timeseries since they were linearly detrended with Climate Data Operator[38]; and the Faithfulness and Causal Markov Condition assumptions which state that two variables that are conditionally independent given a set of other variables are graphically separated given that set in the graph, and vice versa[21]. As discussed earlier, these two assumptions can be deemed reasonable because they imply that the independencies that we measure arise from the causal structure, and not from a fine-tuning of parameters underlying the data generating process[42]. Lastly, causal sufficiency assumption assumes the graph includes all the common causes. While one cannot ignore the fact that Sea Surface Temperature (SST) interacts with other variables in the climate system, it is important to acknowledge that temperature plays a fundamental role in climate modeling. It is explicitly considered in the equations of conservation of mass, energy, and momentum, which are implemented in climate models. Hence, it remains reasonable to approximate SST as part of a closed system.

The analysis pipeline is as follows: we utilize the causal discovery algorithm PCMCI for inferring the links from the set of nodes detected in the reference dataset HadISST. We feed the algorithm with the thirty temporal signals inferred with δ-MAPS. The significance level α and the maximum lag $\tau_{max}$ are set to 0.05 and 3, respectively. The sensitivity of the detection of links to α is shown on Supplementary Table 2. The first step of PCMCI is a condition-selection step (PC1) to learn the sets of parents $P(X_t^j)$ for all variables $X_t^j \in X_t$. In these sets all adjacencies due to indirect paths or common drivers are removed. The second step is the Momentary Conditional Independence (MCI) test for testing $X_{t-\tau}^i \to X_t^j$:

$$\text{MCI}: X_{t-\tau}^i \not\!\perp X_t^j \mid \widehat{\mathscr{P}}(X_t^j) \setminus \{X_{t-\tau}^i\}, \widehat{\mathscr{P}}(X_{t-\tau}^i) \qquad (2)$$

This test has been shown to work well for highly autocorrelated time series[32]. Here we approximate the dependencies as linear

and choose Partial Correlation to perform the conditional independence tests.

## Metrics

Two distance metrics $WWD$ and $D_{ACE}$ were designed to evaluate different structural aspects of the networks. Regions and the causal structure have first been inferred in the HadISST dataset (reference network[26]), and $WWD$ and $D_{ACE}$ evaluate subsequently the simulation datasets through the SST signals and causal effects associated to domains and links. If readers are interested in evaluating the inference of regions and links itself, two additional metrics, $D_{NMI}$ and $D_{F_1}$, are presented in the Supplementary Note 3.

## Weighted Wasserstein distance

$WWD$ focuses on variability at regional scale by comparing the SST distributions in regions with respect to the referent SST distributions. $WWD$ is based on the Wasserstein Distance ($WD$) metric, that comes from optimal transport theory and gives distances between two distributions[43]. It has an intuitive interpretation as it is the amount of work required to transform one probability distribution into another, and is retrieved with the scikit learn library[44]. Regions are weighted because they do not play the same role in the climate system (Fig. 2b). Here we use the notion of strength ($S$) of a region, that is for a given region the sum of the co-variances it shares with all the other regions of the network[24]. For a region associated to a signal of temperature $T_{sim}(t)$, in a dataset made of $n$ regions, its strength is expresses as $S_i = \sum_{\substack{j=1 \\ j \neq i}}^{n} Cov(T_i(t), T_j(t))$ and reflects its contribution to the total variability of the system. The strengths of regions are directly provided by δ-MAPS algorithm. The weight of the region then corresponds to the proportion of its strength relative to the total sum of strengths, expressed as $p_i = S_i / \sum S$.

Finally, $WWD$ between two sets of signals extracted from the regions of the reference and the regions of the simulation and noted respectively $= \{T_{ref_1}, \ldots, T_{ref_n}\}$ and $T_{sim} = \{T_{sim_1}, \ldots, T_{sim_n}\}$.

$$WWD(sim, ref) = \sum_{i=1}^{n} p_i WD\left(T_{ref_i}, T_{sim_i}\right) \qquad (3)$$

## Distance average causal effect

Distance Average Causal Effect ($D_{ACE}$) evaluates the regions' influences as causal gateways−or causal drivers−in the system. As mentioned earlier, we assume a linear model based on the reconstructed time series graph, and we estimate the linear causal effect (CE) of perturbations[31]. These linear causal effects are estimated with the Linear Mediation class of Tigramite, using the Wright approach[45]. They correspond to normalized measures of the strength of the causal dependencies. A hypothetical propagation in a causal network may propagate along direct and indirect causal paths. There is one path coefficient for a direct causal path and several path coefficients for an indirect causal path. For a causal link $X_{t-\tau}^i \to X_t^j$, the path coefficient is defined as the standardized regression coefficient between the components $X_t^j$ and $X_{t-\tau}^i$, and representing the expected change of $X^j$ at time $t$ if $X^i$ was perturbed at time $t - \tau$ by one standard deviation. Unlike co-variances, causal effects are asymmetric measures and follow the direction of the arrows. The total causal effect between two regions $i$ and $j$ at a specific lag $\tau$ (measured in months) is denoted $I_{i \to j}^{CE}(\tau)$ and is computed as the sum of the product of the causal effects along each causal path between $i$ and $j$[31]. All causal paths between two regions are included in the measure of the total causal effect, and it is possible that causal effects along the different paths have different signs. By extension, the Average Causal effect ($ACE$) is defined as the average of the outgoing total causal effects of a given node, and indicates how important is a region as driver of the variability in the

system. $D_{ACE}$ calculates the distance between the spatial grid of the simulation $M_{sim}$ and the spatial grid of the reference $M_{ref}$, in which grid cells are labeled with the value of the $ACE$ of the region it belongs to. The grid cells in the map of the simulation and in the reference map are also randomly permuted, and the distance between these permuted maps (noted $M_{sim'}$ and $M_{ref'}$) is also calculated to serve as denominator (Eq. (4)). Values of $D_{ACE}$ near 0 indicate a good agreement between the $ACE$ assigned to regions and the truth values, while high values of metric $D_{ACE}$ indicate the $ACE$ values assigned to the regions are close to the ones obtained by chance.

$$D_{ACE}(sim, obs) = \frac{\sum_{i=1}^{n}|M_{sim}(i) - M_{ref}(i)|}{\sum_{i=1}^{n}|M'_{sim}(i) - M'_{ref}(i)|} \quad (4)$$

A similar metric has been proposed in Falasca[24]. Overall, $D_{ACE}$ is a very comprehensive metric to evaluate the connections, and tell us how well a perturbation propagates in the SST system with respect to reference.

## Data availability
The reference datasets HadISST and COBEv2 can be obtained respectively at https://www.metoffice.gov.uk/hadobs/hadisst/data/download.html and https://psl.noaa.gov/data/gridded/data.cobe2.html. The CMIP6 outputs can be obtained at https://esgf-node.llnl.gov/projects/cmip6/.

## Code availability
The δ-MAPS software can be obtained in https://github.com/FabriFalasca/delta-MAPS. The PCMCI software and python scripts to estimate causal effects are available online under https://github.com/jakobrunge/tigramite. The python scripts to pre-process the datasets, to compute the metrics and to derive climate sensitivity estimates are available at https://github.com/ricardlu /netCS.

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

## Acknowledgements

This research was supported by the European Union's Horizon 2020 research and innovation program under Marie Skłodowska-Curie grant agreement No. 860100 (iMIRACLI), by the FORCeS project under the European Union's Horizon 2020 research program with grant agreement No. 821205, and by the CleanCloud project under the Horizon Europe research program with grant agreement No. 101137639.

## Author contributions

A.N. suggested and designed the study. F.F. and J.R. contributed to application of methods. L.R. led the scientific analysis and paper writing. All authors (i.e. L.R., F.F., J.R. and A.N.) contributed to the scientific interpretation of the results and to the paper writing.

## Competing interests

The authors declare no competing interests.
