## [Peer Review File · Nature Communications]

Network-based constraint to evaluate Climate SensitivityREVIEWER COMMENTS

Reviewer #1 (Remarks to the Author)

netEC: an emergent constraint on climate sensitivity based on network analysis. L. Ricard, F. Falasca, J. Runge, and A. Nenes.

Recommendation: Publication not recommended without major revision.

The paper presents an analysis method for sea surface temperature (SST) that reduces the dimensionality of spatial patterns into regions of correlated SST and the relationships among these regions on time scales up to 3 months. Modeled and observed analyses can be compared to evaluate model performance. Two measures are introduced to compare the fidelity of the patterns (normalized weighted Wasserstein distance, *NWWD*) and pattern dependencies on each other (distance average causal effect, *DACE*) to analyzed observations. The method is novel and an important complement to other comparison metrics, like bias, root mean square error, and correlation coefficients.

The skill with which models produce the regional patterns and their dependencies on one another is presented as an emergent constraint to weight model equilibrium climate sensitivity (ECS) and transient climate response (TCR). However, the new skill metrics presented in the paper do not constitute an emergent constraint. An emergent constraint, as the authors correctly summarize in ll. 29-33, is an in-model relationship between a historical or present-day, observable model property and a future-climate property of that model. *NWWD* and *DACE* are not shown in the paper to be related to in-model ECS or TCR. Thus, though *NWWD* and *DACE* applied to SST patterns are important and novel new metrics, the paper does not establish them as emergent constraints. For this reason, I recommend against publication of the paper in its present form.

The new metrics merit publication, and using them to weight models is of interest. The authors could also pursue their research further by examining whether model network properties are correlated to ECS and TCR within the models. If such a relationship were demonstrated, a new emergent constraint would indeed be identified.

Some additional suggested revisions follow.

Major Issues

1. ll. 131-146: As general support for using the metrics developed in this paper to weight models, the authors claim that models that reproduce better past periods embed a better set of physics (l. 141). This assertion is unfounded for two reasons: (1) Experience by major modeling centers does not show that improving the physical basis of models immediately translates into better simulation of past climates. The experience of multiple modeling centers upon initial inclusion of cloud-aerosol interactions is an example. In many cases the simulation of historical temperature trends worsened, a consequence of the uncertainties associated with modifying large cloud radiative effects in the models. Yet, few would argue earlier physics which completely neglected aerosol-cloud interactions in these models was “better,” even if historical temperature simulations were. (2) As the authors note on l. 39, the possibility exists that models are tuned to simulate past and present behavior. Better historical simulations may reflect tuning, rather than better physics. This applies both to historical simulations and patterns, as bias, root mean square, and correlation coefficients between models and observations receive significant attention in model development.

2. ll. 172-176: Regarding the comparison with the IPCC AR6 sensitivity assessments, the IPCC assessments largely follow Sherwood et al. (2020, *Rev. Geophys.*), which combine multiple lines of evidence using a Bayesian approach. For this reason, the uncertainty narrows relative to any single estimate. Were the analysis methods developed in this paper to become the basis for an emergent constraint they could be combined with other lines of evidence and possibly narrow the uncertainty range. The discussion in ll. 35-38 also requires revision. The referenced studies (5,10,11) appear to rely on considerably less evidence than the IPCC AR6 sensitivity estimates. A broader summary of current ECS/TCR estimates is required, including at least references to IPCC AR6 and placing the referenced studies in the context of these more broadly based estimates.

3. In physical models of the Earth system, “causal” often refers to a mechanism that can be traced to or emerges from established dynamic, thermodynamic, or chemical principles. Here, it refers to autocorrelated changes in different regions. These are statistical relationships. Although this is clear from context, the distinction is probably worth noting.

4. ll. 78-79: IPCC AR6 estimates the effective radiative forcing from aerosols to lie between -1.9 and -0.2 W m^{-2} for the period 1750 to 2019. Total anthropogenic

forcing is estimated at 2.7 W m^{-2} . The aerosol forcing is not small to a high degree of certainty.

Minor Points

1. Reference 1 is incomplete, listing only author, title, and year.
 2. l. 188: Is this really a best estimate in anything other a statistical sense, given the lowest and highest values are weighted most heavily?
 3. Many models consider bias, root mean square error, and correlation relative to observations heavily in their development process. The latter measures capture information about regional model fidelity. Is there any overlap between these measures and *NWWD*?
 4. l. 271: Provide explanation of how this detrending was done?
 5. Indicate lag units (months) on Fig. 3d.
 6. ll. 151-156: This text is very important to understanding the weighting but requires going to the supplementary material to understand well. A modest expansion of the details here could be helpful to readers.
- l. 198, “does” – > “do”

Reviewer #2 (Remarks to the Author)

Review: NCOMMS-23-42799

Title: netRC: an emergent constraint on Climate Sensitivity based on network analysis.

Authors: Ricard, L., F. Falasca, J. Runge, and A. Nenes

Overview and major comments

The present paper uses an advanced causal network framework to quantify the ability of CMIP6 models to represent the climate dynamical system, and thereby constrain the Equilibrium Climate Sensitivity (ECS) and Transient Climate Response (TCR). The authors focus on Sea Surface Temperatures (SST). Starting from gridded datasets, either from observations or model output, they use the δ -MAPS approach to reduce the dimension of the problem and identify geographical domains for which grid cells coherently covary. Then, they perform causal inference based on the PCMCI algorithm to identify the causal patterns that describe the dynamical interactions between these domains. The causal network obtained with SST observations serves as a reference for evaluating CMIP6 models. Two metrics are introduced, namely the Normalized Weighted Wasserstein Distance (*NWWD*) and the Distance Average Causal Effect $_{DACE}$. *NWWD* quantifies the model skill to capture intra-domain SST distributions (aggregated over all domains), while $_{DACE}$ quantifies the model ability to reproduce the observed causal dependencies between domains. Finally, these metrics are combined to weight CMIP6 models according to their performance and thereby constrain the distribution of ECS and TCR.

The paper is interesting, logically structured and overall well written. The statistical approach to quantify model performance appears rather novel and original, at least in the context of constraining ECS/TCR. Nevertheless, I have a few major comments that I think need to be addressed before the paper can be considered for publication.

1. The authors state that they are developing an 'emergent constraint' on ECS and TCR. Following the review of Williamson et al. (2021), I think the author should revise their vocabulary and position. One important aspect in developing an emergent constraint is to provide a physical understanding on how the present-day metric is relevant to

future climate changes. Except a quick mention of the Fluctuation-Dissipation Theorem (FDT), I did not find any physical insight on how *NWWD* and *DACE* metrics are truly relevant for ECS and TCR. A minimum in identifying an 'emergent constrain' would at least be to plot ECS (or TCR) as a function of *NWWD* or *DACE* and thereby quantifying the functional dependency between these two variables. Besides, a more detail argument to properly invoke the FDT is required, following the discussion of Williamson et al. (2021). Note also that the *NWWD* and *DACE* metrics are profoundly different from what is usually used as a present-day metric in an 'emergent constraint' analysis (see e.g., Table 1 in Williamson et al. 2021), at least because they are performance metrics, not metrics quantifying a behavior or feedback of the climate system. This should be discussed as well.

2. It remains difficult to interpret the *NWWD* and *DACE* metrics: because it mixes time and spatial variability, what does *NWWD* truly quantify? Is it dominated by time or spatial variability? *NWWD* aggregates performance over 30 domains, but is *NWWD* dominated by only a few regions (e.g., in relationship with Fig 3b)? Or are some regions more critical in 'constraining' ECS/TCR? Does *NWWD* inter-model variability depend on only a few regions? Below which value would you say a model is close to observations (maybe given its internal variability and given observational uncertainties)? Similar questions do hold for *DACE*.
3. For a given model, the sensitivity of *NWWD* and or *DACE* can be really high to model internal variability or to observational uncertainty (HadISST vs. COBEv2, see Figure 1 in the supplementary material and the short discussion p4, l131-134). These sources of uncertainty do not seem to be accounted properly when constraining ECS and TCR. For internal variability, this remains unclear how each model ensemble is treated. Are the metrics computed over each ensemble member before being averaged? For observational uncertainty, this looks rather inappropriate to just consider the average of the metrics computed with each reference dataset (especially in an 'emergent constrain' analysis, and because they yield rather different results for *DACE*) and not to consider the obtained range and thereby more appropriately quantify model behavior.

More minor comments

1. p2, l24-25: unclear if your statement concerns the ECS distribution within the CMIP5 and CMIP6 ensembles or some median or average

ECS.

2. p2, l27-28: such a statement requires a reference or further argument/discussion.
3. p2, l36: what do mean by “currently expected”?
4. p2, l39: “reasonably well reproduced” sounds rather vague.
5. p2, l39-40: note Hourdin et al. (2017) discuss that the global mean temperature is a result of model tuning, not a priori its past trend. The use of temperature past trend as a tuning metric varies from one modeling center to another.
6. p2, l41-42: unclear what you mean with these “large deviation of SST”. Do you mean SST pattern? Besides, the use of the word “bias” for ECS/TCR looks inappropriate.
7. p2, l47-51: these few sentences are critical for justifying part of your approach. These look rather like some subjective claims not sufficiently argued and discussed (see also major comments). E.g, to what extent is the FDT applicable to your framework? How can you state that “the behavior of the network is a good proxy of the behavior of the model under an increase of CO₂”? Why would it make the emergent constrain “more robust”? To what extent your causal predictors – and the way you quantify them – are relevant for climate change? Of course, it might be hard to have a fully convincing argument, but at least the limitations should be clearly discussed.
8. p3, l58: the PCMC acronym is not provided.
9. p3, l75-76: “In order to...” It really hard to understand this sentence.
10. p3, l77-78: Zelinka et al. (2020) show that the CMIP6 ECS increase is primarily related to extratropical low clouds. Though important, low clouds in the tropics and subtropics are not the only source of ECS/TCR uncertainty.
11. p3, l82-83: “merge regions that covary over time”. This is a bit confusing. Do you mean grid cell?
12. p3, l87-88: I did not fully understand this sentence.
13. p3, l88-89, Figure 3c: you mention “SST time series at regional scales”. Do you mean time series of region-average (or domain-average) SST? Note it is not always clear if you make a difference between “region” and “domain”. “domain” seems related to what you obtain from the d-MAPS dimension reduction algorithm. Besides, Figure 3c provides

the time series distribution, not the time series itself as I was expected when reading the sentence. You call it “ENSO signal”. Have you quantified to what extent it truly correlates with more standard ENSO time series?

4. p4, l108: are your results sensitive to this choice of 3 months as a maximum time lag?
5. p4, l118: I found “past performance” quite confusing. I would rather use something like CMIP6 model performance over the historical period.
6. p4, l125-126: when you write “global”, do you mean tropical/subtropical or truly global? Besides, to some extent, the two metrics combines the same data, but there is some geographical weighting in *NWWD*. Are there regions that dominates *NWWD* values and its inter-model spread?
7. p4, l132-133: I feel “mean” is inappropriate.
8. p4, l133-136: the correlation between HadISST-based and COBEv2-based *DACE* is weak. This is a concern for using *DACE*. How different are the HadISST and COBEv2 causal network? How do you understand those differences? Is the use of the average between HadISST-based and COBEv2-based metrics appropriate? Why not keeping this source of uncertainty in your subsequent analysis?
9. p4, l141-146: this statement is not obvious to me. Error compensations may help models to capture the past period, but these error compensations may reveal critical for the climate response to increased CO₂. Besides, the subsequent argument, based on Runge et al. (2019), who only formulated it, looks rather weak. However, this is truly important for justifying the used of *NWWD* and *DACE* to constrain ECS and TCR.
10. p5, 154: does those value of σD lead to aggressive weighting?
11. p5, l170-172: This is an important point when considering the likely ranges of ECS and TCR you are provided in this paper, especially in the abstract. I feel this requires further discussion and analysis, so that these likely ranges can be truly relevant to the climate community and beyond. Otherwise, admit that your paper is mostly methodological, and that your obtained ranges will need further assessment and more consistent comparison with previous studies.
12. p6, l198-199: this is too vague as a recommendation. Can you already identify from the literature another emergent constrain that may sep-

arate low- and high-sensitivity models? And if yes, to what extent would your own constraint still be of any interest?

0. p6, l206: “interpretable metrics”. This is an important point, especially within the context of ‘emergent constraints’. However, I did not find it is sufficiently addressed in the previous sections (in terms of physical interpretation).
1. p6, l229: not sure “global mean” is appropriate.
2. p6, l231: 1975-2015 corresponds to 41 years, thus 492 months. Am I missing something? This would help to talk about ‘time samples’.
3. p8, l323-324: I was not able to understand how you compute this total causal effect between two domains.
4. p9, l329: “randomly permuted maps”. It is unclear how you use them. Do you perform some kind of iteration, as in a bootstrap approach?

Reference

- Hourdin, F. *et al.* The Art and Science of Climate Model Tuning. *Bull. Am. Meteorol. Soc.* 98, 589–602 (2017).
- Runge, J. *et al.* Inferring causation from time series in Earth system sciences. *Nat. Commun.* 10, 2553 (2019).
- Williamson, M. S. *et al.* Emergent constraints on climate sensitivities. *Rev. Mod. Phys.* 93, 025004 (2021).
- Zelinka, M. D. *et al.* Causes of Higher Climate Sensitivity in CMIP6 Models. *Geophys. Res. Lett.* 47, e2019GL085782 (2020).

Response to Anonymous Referee #1, NCOMMS-23-42799 reviews

We would like to thank Anonymous Referee #1 for their constructive comments and thorough efforts in helping to improve the quality of the paper. We took consideration of their comments and remarks and modified the manuscript as detailed below.

Comment:

The paper presents an analysis method for sea surface temperature (SST) that reduces the dimensionality of spatial patterns into regions of correlated SST and the relationships among these regions on time scales up to 3 months. Modeled and observed analyses can be compared to evaluate model performance. Two measures are introduced to compare the fidelity of the patterns (normalized weighted Wasserstein distance, NWWD) and pattern dependencies on each other (distance average causal effect, D_{ACE}) to analyzed observations. The method is novel and an important complement to other comparison metrics, like bias, root mean square error, and correlation coefficients.

The skill with which models produce the regional patterns and their dependencies on one another is presented as an emergent constraint to weight model equilibrium climate sensitivity (ECS) and transient climate response (TCR). However, the new skill metrics presented in the paper do not constitute an emergent constraint. An emergent constraint, as the authors correctly summarize in ll. 29-33, is an in-model relationship between a historical or present-day, observable model property and a future-climate property of that model. NWWD and D_{ACE} are not shown in the paper to be related to in-model ECS or TCR. Thus, though NWWD and D_{ACE} applied to SST patterns are important and novel new metrics, the paper does not establish them as emergent constraints. For this reason, I recommend against publication of the paper in its present form.

The new metrics merit publication, and using them to weight models is of interest. The authors could also pursue their research further by examining whether model network properties are correlated to ECS and TCR within the models. If such a relationship were demonstrated, a new emergent constraint would indeed be identified.

Reply:

We thank the reviewer for raising this comment. Indeed, we do not introduce a traditional emergent constraint, as we do not have an explicit functional form that links the network metrics with the ECS and TCR. We do show however that netEC does constrain implicitly the ECS/TCR as our performance weights have a clear physical link to climate sensitivity. We could make the relationship more explicit (e.g., by tuning the shape parameter σ_D to optimize the correlation between ECS/TCR and the weights) to formulate a “traditional” emergent constraint, but we choose rather to use a calibration approach that ensures there is no overconfidence in models¹. One advantage of weighting estimates over estimates derived from linear regression framework is to avoid the caveat of the p-hacking and overconfidence in models².

In response to the reviewer comment, we now refrain from using the term “emergent constraint” but rather emphasize that the estimates derived from the emergent SST networks should constitute a new and important constraint for ECS and TCR. There are good reasons for maintaining the link with the ECS and TCR. Both distance metrics characterize the deviation of regional SST anomalies to reference ones. In the past years the climate sensitivity estimates derived from climate models were substantially higher from the ones derived from observed warming record and radiative balance³. This discrepancy was due to the time dependence of the radiative feedback to the spatial pattern of SST in models (so called “pattern effect”). The large inter-model spread in ECS estimates is explained by the large inter-model spread in radiative

feedback, which itself is dominated by the spread in model physics at fast time scales and the spread in the magnitude of the pattern effect on slow time scales.⁴ In other words, the climate model feedback is strongly dependent of the evolving patterns of surface warming⁵ (which explains the curvature in the $N-T$ Gregory regression⁶). Moreover, there are non-local connections between the warming pattern and feedbacks via the atmospheric circulation or local non-linearities, and a purely local framework cannot give an entirely adequate explanation⁵. For these reasons, we think our estimates derived from regional SST anomalies and their interactions in CMIP6 models are relevant to estimate ECS and TCR. netEC estimates should constitute a new line of evidence, because our performance weights are not simply “performance metrics” but have a clear physical link to climate sensitivity.

A simple summary of the above then is to say the fully coupled climate models embed our best understanding of the relevant feedbacks⁷. The CMIP models encompass feedbacks of the climate system, so do the emergent networks and their statistics. And, the distance of these emergent networks respect to the networks from the reference datasets are directly related to the behavior (or feedback) of the climate systems.

We revised the introduction to better discuss these points.

Major Issues

Comment:

ll. 131-146: As general support for using the metrics developed in this paper to weight models, the authors claim that models that reproduce better past periods embed a better set of physics (l. 141). This assertion is unfounded for two reasons: (1) Experience by major modeling centers does not show that improving the physical basis of models immediately translates into better simulation of past climates. The experience of multiple modeling centers upon initial inclusion of cloud-aerosol interactions is an example. In many cases the simulation of historical temperature trends worsened, a consequence of the uncertainties associated with modifying large cloud radiative effects in the models. Yet, few would argue earlier physics which completely neglected aerosol-cloud interactions in these models was “better,” even if historical temperature simulations were. (2) As the authors note on l. 39, the possibility exists that models are tuned to simulate past and present behavior. Better historical simulations may reflect tuning, rather than better physics. This applies both to historical simulations and patterns, as bias, root mean square, and correlation coefficients between models and observations receive significant attention in model development.

Reply:

The reviewer raises good points here. Indeed, models that provide the “best” simulations may be models that have either a better set of physics, either a better tuning (although, based on what is said in the paper, tuning has a less prominent role because we focus on the detrended SST patterns), either a combination of both. Here the methodology consists in giving more importance to models that have better skill (or past performance) over subsets of the historical period. We add that the ability of models is evaluated through their emergent properties, which arise from the interactions of multiple scales and physical effects in the model, and not directly a consequence of a single processes or parameterizations⁸. In light of the complexity behind these emergent properties, we do not discriminate models based on their tuning or embedded physics.

P5, 1170-173: “The performance of the models over the historical period is an important step towards the estimate of the ECS and TCR, since models that reproduce better the past period embed a better set of physics or have a better tuning that should make them also represent better the future period.”

Comment:

ll. 172-176: *Regarding the comparison with the IPCC AR6 sensitivity assessments, the IPCC assessments largely follow Sherwood et al. (2020, Rev. Geophys.), which combine multiple lines of evidence using a Bayesian approach. For this reason, the uncertainty narrows relative to any single estimate. Were the analysis methods developed in this paper to become the basis for an emergent constraint they could be combined with other lines of evidence and possibly narrow the uncertainty range. The discussion in ll. 35-38 also requires revision. The referenced studies (5,10,11) appear to rely on considerably less evidence than the IPCC AR6 sensitivity estimates. A broader summary of current ECS/TCR estimates is required, including at least references to IPCC AR6 and placing the referenced studies in the context of these more broadly-based estimates.*

Reply:

We may have limited more than we should have the scope of our literature review in the introduction. This is now revised and we give a more comprehensive overview of the current ECS/TCR lines of evidence in the revised paper:

p2, l35-38: “Various lines of evidence are used to estimate climate sensitivity, including constraints from the instrumental records and variability, constraints from climatology, feedback and models, and constraints based on paleoclimate⁷. These lines of evidence are combined in the assessed ranges of ECS and TCR provided in the IPCC AR6⁹.”

Comment:

In physical models of the Earth system, “causal” often refers to a mechanism that can be traced to or emerges from established dynamic, thermodynamic, or chemical principles. Here, it refers to autocorrelated changes in different regions. These are statistical relationships. Although this is clear from context, the distinction is probably worth noting.

Reply:

The point is well taken. Experiments on real-world systems (that would establish causality) are extremely challenging to carry out¹⁰. However, the large amount of data available compared to randomized controlled trials has opened the way to a new research area where causal effects may be estimated from observational data¹¹. In this data-driven context the term “causal” has been established for a number of years now^{6,7}, and is how we use it here. PCMCi is an appropriate data-driven approach, as the causal associations are detected, to the degree the data allow, directly from the time series data.¹² Judea Pearl made first the distinction very clear between causal inference approach and statistical inference approaches in¹³: “An associational concept is any relationship that can be defined in terms of a joint distribution of observed variables, and a causal concept is any relationship that cannot be defined from the distribution alone”. Later in the text: “associations characterize static conditions, while causal analysis deals with changing conditions”. The term causal in this context has largely been adopted the last few years in the literature.

We have added further clarifications to make it clear to the reader in which context “causality” is used:

p8, l305-307: “The causal discovery algorithm PCMCi (developed in ref¹², implemented in the Tigramite open-source software package for Python and available from <https://github.com/jakobrunge/tigramite>) is a data-driven approach developed to infer causal links and causal effects from large amount of data available.”

Comment:

ll. 78-79: IPCC AR6 estimates the effective radiative forcing from aerosols to lie between -1.9 and -0.2 W m^{-2} for the period 1750 to 2019. Total anthropogenic forcing is estimated at 2.7 W m^{-2} . The aerosol forcing is not small to a high degree of certainty.

Reply:

We absolutely agree here. We say in the text that the uncertainty in aerosol forcing is less now than it was in the past, as a consequence of reduced aerosol forcing. This however does not mean it is insignificant. No changes required here.

Minor Points

Comment:

Reference 1 is incomplete, listing only author, title, and year.

Reply:

Thanks. We corrected it.

Comment:

2. 1. 188: Is this really a best estimate in anything other a statistical sense, given the lowest and highest values are weighted most heavily?

Reply:

True that it is a best estimate in a statistical sense. We replace the term best estimate by the term central estimate, which is more appropriate given the distribution of the weights.

Comment:

3. Many models consider bias, root mean square error, and correlation relative to observations heavily in their development process. The latter measures capture information about regional model fidelity. Is there any overlap between these measures and *NWWD*?

Reply:

The bias, root mean square error and correlation do point-wise comparison at the same time steps, and focus on only certain aspects of the timeseries (the magnitude of the errors and the associations between the observed and simulated time series). Wasserstein Distance is a more comprehensive metric which evaluates the entire distributions, and consider the spatial arrangement of data points.

Comment:

4. 1. 271: Provide explanation of how this detrending was done?

Reply:

We detail in the main paper:

p8, 1305-310: “The causal discovery algorithm PCMCI (...) estimates the causal structure from timeseries corresponding to the nodes of the system, under the following assumptions: stationarity, which can

reasonably be assumed for the SST monthly timeseries since they were linearly detrended with Climate Data Operator¹⁴; (...).”

5. Indicate lag units (months) on Fig. 3d.

Reply:

We modified the legend of Fig.3d:

p16 “Links are labeled with the value of the lag τ in months.”

Comment:

6. ll. 151-156: This text is very important to understanding the weighting but requires going to the supplementary material to understand well. A modest expansion of the details here could be helpful to readers.

Reply:

We explain more in details the calibration approach in the main text:

p5, 1183-191: “The shape parameter is selected within the [20%, 200%] range of the median of the distance metric. For each σ_D value, we assume one by one that a model’s climate sensitivity is the true value and apply a weighting scheme to the remaining models. The σ_D value is retained if 70% of the assumed “true” models fall within the 10-90 percentile range of the weighted distribution of climate sensitivity. The final σ_D value is the minimal value of the possible σ_D values, which provides the most aggressive weighting. We note that the shape parameter is not adjusted to optimize the relationship between the observable and climate sensitivity. Instead, it is tuned to maintain a specific consistency when models are excluded. This calibration approach ensures that we do not instill overconfidence in models, which is a pitfall mentioned in ref². In our study, σ_D values are all comprised between 0.33 and 1.07 (Supplementary Table 3).”

Comment:

7. 198, “does” –> “do”

Reply:

Thanks, corrected.

References

1. Brunner, L. *et al.* Reduced global warming from CMIP6 projections when weighting models by performance and independence. *Earth System Dynamics* **11**, 995–1012 (2020).
2. Williamson, M. S. *et al.* Emergent constraints on climate sensitivities. *Rev. Mod. Phys.* **93**, 025004 (2021).
3. Andrews, M. R., Mark Zelinka, Kristopher B. Karnauskas, Paulo Ceppi, Timothy. Patterns of Surface Warming Matter for Climate Sensitivity. *Eos* <http://eos.org/features/patterns-of-surface-warming-matter-for-climate-sensitivity> (2023).
4. Dong, Y. *et al.* Intermodel Spread in the Pattern Effect and Its Contribution to Climate Sensitivity in CMIP5 and CMIP6 Models. *Journal of Climate* **33**, 7755–7775 (2020).

5. Andrews, T., Gregory, J. M. & Webb, M. J. The Dependence of Radiative Forcing and Feedback on Evolving Patterns of Surface Temperature Change in Climate Models. *Journal of Climate* **28**, 1630–1648 (2015).
6. Gregory, J. M. *et al.* A new method for diagnosing radiative forcing and climate sensitivity. *Geophysical Research Letters* **31**, (2004).
7. Knutti, R., Rugenstein, M. A. A. & Hegerl, G. C. Beyond equilibrium climate sensitivity. *Nature Geosci* **10**, 727–736 (2017).
8. Schmidt, G. A. *et al.* Practice and philosophy of climate model tuning across six US modeling centers. *Geosci. Model Dev.* **10**, 3207–3223 (2017).
9. Calvin, K. *et al.* *IPCC, 2023: Climate Change 2023: Synthesis Report. Contribution of Working Groups I, II and III to the Sixth Assessment Report of the Intergovernmental Panel on Climate Change [Core Writing Team, H. Lee and J. Romero (Eds.)]. IPCC, Geneva, Switzerland.*
<https://www.ipcc.ch/report/ar6/syr/> (2023) doi:10.59327/IPCC/AR6-9789291691647.
10. Runge, J. Causal network reconstruction from time series: From theoretical assumptions to practical estimation. *Chaos: An Interdisciplinary Journal of Nonlinear Science* **28**, 075310 (2018).
11. Yao, L. *et al.* A Survey on Causal Inference. *ACM Trans. Knowl. Discov. Data* **15**, 74:1-74:46 (2021).
12. Runge, J., Nowack, P., Kretschmer, M., Flaxman, S. & Sejdinovic, D. Detecting and quantifying causal associations in large nonlinear time series datasets. *Science Advances* **5**, eaau4996 (2019).
13. Pearl, J. Statistics and causal inference: A review. *Test* **12**, 281–345 (2003).
14. Schulzweida, U., Kornblueh, L. & Quast, R. CDO user guide. *Climate data operators, Version 1*, 205–209 (2006).

Response to Anonymous Referee #2, NCOMMS-23-42799 reviews

We would like to thank Anonymous Referee #2 who has raised important questions concerning key points of the manuscript. We would like to thank them for the work they produced. We took consideration of their comments and remarks and modified the manuscript as detailed below.

Comment:

1) The authors state that they are developing an 'emergent constraint' on ECS and TCR. Following the review of Williamson et al. (2021), I think the author should revise their vocabulary and position. One important aspect in developing an emergent constraint is to provide a physical understanding on how the present-day metric is relevant to future climate changes. Except a quick mention of the Fluctuation-Dissipation Theorem (FDT), I did not find any physical insight on how NNWD and D_{ACE} metrics are truly relevant for ECS and TCR. A minimum in identifying an 'emergent constrain' would at least be to plot ECS (or TCR) as a function of NNWD or D_{ACE} and thereby quantifying the functional dependency between these two variables. Besides, a more detail argument to properly invoke the FDT is required, following the discussion of Williamson et al. (2021). Note also that the NNWD and D_{ACE} metrics are profoundly different from what is usually used as a present-day metric in an 'emergent constraint' analysis (see e.g., Table 1 in Williamson et al. 2021), at least because they are performance metrics, not metrics quantifying a behavior or feedback of the climate system. This should be discussed as well.

Reply:

We agree that we do not introduce a traditional emergent constraint. Identifying a significant linear relationship (and more broadly a functional form between the observable X and the climate sensitivity metric Y) is described in ref¹ as “the most current procedure”, which should open the room to other types of constraints. Here, we do not explicit a functional form between the climate sensitivity and the observables, which makes our constraint implicit. We did not seek to obtain or optimize a correlation (the shape parameter σ_D could be tuned in order to optimize the correlation between ECS/TCR and the weights) but instead used a calibration approach to ensures there is no overconfidence in models². One advantage of weighting estimates over estimates derived from linear regression framework is to avoid the caveat of the p-hacking and overconfidence in models¹.

In response to the reviewer comment, we now refrain from using the term “emergent constraint” but rather emphasize that the estimates derived from the emergent SST networks should constitute a new and important constraint for ECS and TCR. There are good reasons for maintaining the link with the ECS and TCR. Both distance metrics characterize the deviation of regional SST anomalies to reference ones. In the past years the climate sensitivity estimates derived from climate models were substantially higher from the ones derived from observed warming record and radiative balance³. This discrepancy was due to the time dependence of the radiative feedback to the spatial pattern of SST in models (so called “pattern effect”). The large inter-model spread in ECS estimates is explained by the large inter-model spread in radiative feedback, which itself is dominated by the spread in model physics at fast time scales and the spread in the magnitude of the pattern effect on slow time scales.⁴ In other words, the climate model feedback is strongly dependent of the evolving patterns of surface warming⁵ (which explains the curvature in the $N-T$ Gregory regression⁶). Moreover, there are non-local connections between the warming pattern and feedbacks via the atmospheric circulation or local non-linearities, and a purely local framework cannot give an entirely adequate explanation⁵. For these reasons, we think our estimates derived from regional SST anomalies and their interactions in CMIP6 models are relevant to estimate ECS and TCR. netEC estimates should constitute

a new line of evidence, because our performance weights are not simply “performance metrics” but have a clear physical link to climate sensitivity.

A simple summary of the above then is to say the fully coupled climate models embed our best understanding of the relevant feedbacks⁷. The CMIP models encompass feedbacks of the climate system, so do the emergent networks and their statistics. And, the distance of these emergent networks respect to the networks from the reference datasets are directly related to the behavior (or feedback) of the climate systems.

We revised the introduction to better discuss these points.

Comment:

2) It remains difficult to interpret the NWWD and D_{ACE} metrics: because it mixes time and spatial variability, what does NNWD truly quantify? Is it dominated by time or spatial variability? NNWD aggregates performance over 30 domains, but is NNWD dominated by only a few regions (e.g., in relationship with Fig 3b)? Or are some regions more critical in ‘constraining’ ECS/TCR? Does NWWD inter-model variability depend on only a few regions? Below which value would you say a model is close to observations (maybe given its internal variability and given observational uncertainties)? Similar questions do hold for D_{ACE} .

Reply:

That is a good question. The network and its properties emerge from decades of data, from which we remove the SST trend. Both metrics only take as input the SST signals (time series of cumulative SST anomalies), and these signals are always reconstructed from the same regions, i.e. from the exact same number of grid cells. Regions were inferred so that SST grid cells in regions are interconnected. This means there is a guarantee of consistency in the spatial dimension. In other words, there is a control on the fluctuation in space, especially in the reference HadISST dataset (there is at least a moderate correlation between grid cells). In case the SST grid cells in one region are not as much interconnected as the SST grid cells in the reference region, then the time series of these SST grid cells will be associated to various temporal fluctuations, which will be reflected in the SST signals.

To summarize, differences in spatial consistency within regions may be one reason which explains that the temporal fluctuations of the SST signals are different from the ones in the reference SST signals.

Comment:

3) For a given model, the sensitivity of NWWD and or D_{ACE} can be really high to model internal variability or to observational uncertainty (HadISST vs. COBEv2, see Figure 1 in the supplementary material and the short discussion p4, 1131-134). These sources of uncertainty do not seem to be accounted properly when constraining ECS and TCR. For internal variability, this remains unclear how each model ensemble is treated. Are the metrics computed over each ensemble member before being averaged? For observational uncertainty, this looks rather inappropriate to just consider the average of the metrics computed with each reference dataset (especially in an ‘emergent constrain’ analysis, and because they yield rather different results for D_{ACE}) and not to consider the obtained range and thereby more appropriately quantify model behavior.

Reply:

We use all available ensemble members, which results in a total of 300 members from 27 models. The WWD and D_{ACE} of a model are the average values of the WWD and D_{ACE} of its ensemble members. On the boxplots, we only represented these averaged distance metrics.

Indeed, this is important to reduce internal variability and observational uncertainty of metric D_{ACE} . In the submitted paper, the ACE values were calculated along links of a 30-nodes network, where ACE values are all dependent of each other as they are inferred for a specified causal graph. D_{ACE} was based on hundreds of interconnected measures, conversely to WWD which is based on dozens of independent measures of WD in regions. We modified the way to compute the D_{ACE} metric in our revised paper: we consider a reduced SST network to discover the causal links and subsequently quantify the causal effects. The motivation is to focus on the main teleconnections of the climate system i.e. to focus on the teleconnections between climatically-relevant regions. The revised metric D_{ACE} is now based on a SST network reconstructed of only nine regions, which are more climatically-relevant. This allows to gain robustness and interpretability, as D_{ACE} is derived from much less teleconnections.

There is now a very strong agreement between D_{ACE} with respect with HadISST dataset and with respect with COBEv2 dataset (correlation coefficient is equal to 0.97). As there is a perfect agreement for WWD metric, which was already presented in the submitted paper, we think it is appropriate to average the metric values with respect to the two reference datasets.

Intra-model mean (dot) and standard deviation (whiskers), and inter-model mean and standard-deviation (solid and dashed lines) for WWD values (a) and D_{ACE} values (b). The spread of values within models is higher than the spread between models for respectively five and eight CMIP6 models.

More minor comments

- 1) p2, 124-25: unclear if your statement concerns the ECS distribution within the CMIP5 and CMIP6 ensembles or some median or average ECS.

Reply:

We correct in the main text:

p2, 123-25: “ECS distribution derived from CMIP6 ensemble is larger compared to the one derived from CMIP5 ensemble, with the upper bound of ECS distribution shifting towards higher values.”

2) p2, 127-28: such a statement requires a reference or further argument/discussion.

Reply:

We mean that the policy decisions in response to global warming are linked to climate sensitivity estimates. The latest Summary for Policymakers of the IPCC highlights the risks are increasing with every increment of warming⁸. In turn, mitigation and adaptation pathways are aligned with the warming scenarios.

3) p2, 136: what do mean by “currently expected”?

Reply:

We wanted to refer the assessed range of climate sensitivity estimates of the latest IPCC report. We correct in the main text:

p2, 140-41: “These studies conclude that the Earth’s surface will warm less than currently expected in ref⁸”

Comment:

4) p2, 139: “reasonably well reproduced” sounds rather vague

Reply:

Yes, it is vague. We modify the sentence as well as the reference:

p2, 142-46: “SST is one of the most important of climatology variables that models need to capture to reproduce the observations for the 20th century warming⁹. While the confidence in the historical trajectory of global mean temperatures is high in models, the confidence in regional scale SST variability is lower and there are large discrepancies between the observed and modelled SST trends in warming patterns¹⁰”

Comment:

5) p2, 139-40: note Hourdin et al. (2017) discuss that the global mean temperature is a result of model tuning, not a priori its past trend. The use of temperature past trend as a tuning metric varies from one modeling center to another.

Reply:

We addressed the comment with the previous comment.

Comment:

6) p2, 141-42: unclear what you mean with these “large deviation of SST”. Do you mean SST pattern? Besides, the use of the word “bias” for ECS/TCR looks inappropriate.

Reply:

Indeed, the term deviations is inappropriate. We correct in the main text:

p2, 144-46: “While the confidence in the historical trajectory of global mean temperatures is high in models, the confidence in regional scale SST variability is lower and there are large discrepancies between the observed and modelled SST trends in warming patterns¹⁰”

Comment:

- 7) p2, 147-51: these few sentences are critical for justifying part of your approach. These look rather like some subjective claims not sufficiently argued and discussed (see also major comments). E.g., to what extent is the FDT applicable to your framework? How can you state that “the behavior of the network is a good proxy of the behavior of the model under an increase of CO2”? Why would it make the emergent constrain “more robust”? To what extent your causal predictors – and the way you quantify them – are relevant for climate change? Of course, it might be hard to have a fully convincing argument, but at least the limitations should be clearly discussed.

Reply:

We agreed we needed to strengthen the link between climate sensitivity and network properties. We discussed already it in our response to the first major issue raised.

In the way we construct the network, we know the response of the network properties to an increase of CO2 is a proxy of the response of the regional SST anomalies to CO2 increases. It has already been shown that the behavior of a model under an increase of CO2 is dependent on the warming patterns at slow time scales⁴. By extension, the network properties are relevant for climate change.

One possible limitation is that the network properties represent model behavior of the models for the period of time considered, here selected as the longest possible period during with least uncertain aerosol forcing (1975-2014).

Comment:

- 8) p3, 158: the PCMCI acronym is not provided.

Reply:

We provide the acronym in the main text:

p3,169-72: “As a second step, causal links are identified with causal discovery algorithm PCMCI, which is an adaptation of the condition-selection PC algorithm (named after its inventors Peters Spirtes and Clark Glymour) followed by the Momentary Conditional Independence (MCI) test¹¹.”

Comment:

- 9) p3, 175-76: “In order to...” It really hard to understand this sentence.

Reply:

We rephrase in the main text:

p3, 189-90: “We constrain the distributions of climate sensitivity with properties of emergent networks, reconstructed from SST anomalies from global simulations.”

Comment:

10) p3, 177-78: Zelinka et al. (2020) show that the CMIP6 ECS increase is primarily related to extratropical low clouds. Though important, low clouds in the tropics and subtropics are not the only source of ECS/TCR uncertainty.

Reply:

We agree, and remove the sentence.

11) p3, 182-83: “merge regions that covary over time”. This is a bit confusing. Do you mean grid cell?

Reply:

Yes, we mean grid cells. We correct this in the main paper:

p3, 194-96: “Grid cell temperatures are not independent from their neighbor values; therefore, it makes sense to merge grid cells that covary over time into regions that have a specific role in the time-evolving climate system.”

Comment:

12) p3, 187-88: I did not fully understand this sentence.

We modify the sentence as follows:

p4, 199-102: “We can associate each SST region to its SST signal, which is defined as the area-weighted cumulative SST anomalies of the grid cells of the regions (further discussed in Methods section). Grid cells are weighted here with the cosine of their latitude.”

Reply:

Comment:

13) p3, 188-89, Figure 3c: you mention “SST time series at regional scales”. Do you mean time series of region-average (or domain-average) SST? Note it is not always clear if you make a difference between “region” and “domain”. “domain” seems related to what you obtain from the δ -MAPS dimension reduction algorithm. Besides, Figure 3c provides the time series distribution, not the time series itself as I was expected when reading the sentence. You call it “ENSO signal”. Have you quantified to what extent it truly correlates with more standard ENSO time series?

Reply:

We show below the time serie of the “ENSO signal” obtained in the HadISST dataset, over the period 1975-2014. We notably see the strong events of the years 1982-1983 and 1997-1998.

Time serie of the “ENSO signal” in the HadISST dataset.

14) p4, 1108: are your results sensitive to this choice of 3 months as a maximum time lag?

Reply:

Yes, the metric D_{ACE} is sensitive to the choice of the maximum time lag. There is a moderate correlation between our D_{ACE} values ($\tau_{max} = 3$ months) and D_{ACE} values ($\tau_{max} = 12$ months), with Pearson correlation coefficient equals to 0.42.

The causal effect of on region on another region is measured for each time lag within the range of time lag specified. But the Average Causal Effect relies on the maximum causal effect among the causal effects measured within the range of time lags. This explains such sensitivity of D_{ACE} to the time lag.

Comment:

15) p4, 1118: I found “past performance” quite confusing. I would rather use something like CMIP6 model performance over the historical period.

Reply:

We accept the suggestion, and rename the subsection p4, 1144: “CMIP6 model performance over the historical period”.

Comment:

16) p4, 1125-126: when you write “global”, do you mean tropical/subtropical or truly global? Besides, to some extent, the two metrics combines the same data, but there is some geographical weighting in $NWWD$. Are there regions that dominates $NWWD$ values and its inter-model spread?

Reply:

This is a good question. Yes, the WWD values of models are largely dominated by some regions. We do not expect regions to play equivalent role in the climate regions. The inference of regions is done on the criterion of the consistency of SST grid cells, not on their relevance for the climate. We notably expect the big regions inferred in the tropics to be more climatically relevant, like the ENSO region, Indo-Pacific Warm Pool region, Indian Ocean region, Southern Pacific Ocean etc.

We show below the cumulative sum of the proportion of strengths of the regions. We recall the strength of a region is a correlation-based measure, which is equal to the sum of the absolute correlation coefficient of a region with all other regions. Half of the regions don’t contribute to WWD . The ENSO region contributes to one third of the WWD values. We add this figure in the supplementary information.

Cumulative sum of the proportion of strengths as function of regions. Nine regions contribute to 90% of the WWD.

Comment:

17) p4, 1132-133: I feel “mean” is inappropriate.

Reply:

We remove the term “mean”.

Comment:

18) p4, 1133-136: the correlation between HadISST-based and COBEv2- based D_{ACE} is weak. This is a concern for using D_{ACE} . How different are the HadISST and COBEv2 causal network? How do you understand those differences? Is the use of the average between HadISST-based and COBEv2-based metrics appropriate? Why not keeping this source of uncertainty in your subsequent analysis?

Reply:

This is a very good and important point. Yes, we can explain the differences. On one hand, WWD values are based on a subset of regions (see our response to comment 16) above), and the WD values are independent of each other. On the other hand, ACE values are calculated along links of a 30-nodes network, and are all dependent of each other as they are inferred for a specified causal graph. D_{ACE} is based on hundreds of interconnected measures, while WWD is mostly based on ten independent measures. This explains the high sensitivity of the D_{ACE} metric compared to the WWD metric, and we modified the D_{ACE} metric to reduce this uncertainty in our revised paper (see summary of revisions and the revised manuscript and SI).

In our submitted paper, as well as in our revised paper, there is a perfect correlation between the WWD values with respect HadISST and with respect to COBEv2 ($\rho = 1.00$). Also, the absolute difference $|WWD_{HadISST} - WWD_{COBEv2}|$ is smaller than the inter-model spread for all models.

In our revised paper, D_{ACE} is based on a subset of SST regions (and a fortiori a subset of teleconnections). There is now a very strong agreement between the values with respect with HadISST and the values with

respect to COBEv2 ($\rho = 0.97$). Also, the absolute difference $|D_{ACE_{HadISST}} - D_{ACE_{COBEv2}}|$ is smaller than the inter-model spread for all models.

We show below the ECS and TCR estimates derived from metrics with respect to HadISST only, with respect to COBEv2 only, and with respect to both HadISST and COBEv2. We add this figure in the supplementary information.

ECS estimates (a) and TCR estimates (b) with weights derived from distance to HadISST dataset (left), distance to COBEv2 dataset (center) and distance to both HadISST and COBEv2 datasets (right).

Comment:

19) p4, 1141-146: this statement is not obvious to me. Error compensations may help models to capture the past period, but these error compensations may reveal critical for the climate response to increased CO₂. Besides, the subsequent argument, based on Runge et al. (2019), who only formulated it, looks rather weak. However, this is truly important for justifying the used of *NWWD* and D_{ACE} to constrain ECS and TCR.

Reply:

We discussed the relevance of metrics *WWD* and D_{ACE} to constrain climate sensitivity above, in the response to major comment.

Comment:

20) p5, 154: does those value of σ_D lead to aggressive weighting?

Reply:

σ_D can take values comprised between 0.20 and 2. In the revised paper, three out of four σ_D have low values, which may be qualified of aggressive weighting. That being said a same σ_D value leads to different weights depending on the distribution of the distance metric values. Best way to describe the weighting is to visualize

the trajectory of the performance weights. In our case, around 10 CMIP6 out of the 27 models are negligible in the finale estimate of ECS. We add this figure in the supplementary information.

Performance weights of ECS (blue) and TCR (red) as function of the CMIP6 models, where models are sorted in decreasing order. Weights are derived from exact same distance metrics values, but the shape parameters determined with the calibration approach lead to a more aggressive weighting for ECS, for which a model contributes to more than 25% of the final estimates. We see 11 models are discarded for ECS estimates, with negligible weights.

Comment:

21) p5, 1170-172: This is an important point when considering the likely ranges of ECS and TCR you are provided in this paper, especially in the abstract. I feel this requires further discussion and analysis, so that these likely ranges can be truly relevant to the climate community and beyond. Otherwise, admit that your paper is mostly methodological and that your obtained ranges will need further assessment and more consistent comparison with previous studies.

Reply:

Our study is based on the analysis of 27 CMIP6 models. The more models are included, the more likely it is to start with a large initial (“unweighted”) distribution. We just ask here to interpret with caution our weighted estimates, which can solely be compared to the unweighted estimates. This does not question the methodology nor the objective of the paper. We agree that the ultimate goal would be to put into perspective our constrained range (as every constrained range published). Combining lines of evidence is really challenging, owing to the different nature of the data and methodology used.

No changes required here.

Comment:

22) p6, 1198-199: this is too vague as a recommendation. Can you already identify from the literature another emergent constrain that may separate low- and high-sensitivity models? And if yes, to what extent would your own constraint still be of any interest?

Reply:

The term “clusters” does not appear in the Williamson et al. review (2021).

When it comes to understand the origin of the inter-model differences in climate sensitivity estimates, studies have grouped models depending on how well some processes are reproduced in them. Back in 2005, Bony and Dufresne already separated high- and low-sensitivity models depending on the radiative response of tropical cloud to global warming¹² by investigating the inter-model differences in tropical cloud feedback. However, to produce climate sensitivity estimates, there is often a significant statistical relationship between one observable and the climate sensitivity metrics, which a fortiori gives right to low- or high-sensitivity models, depending if the final estimate is more towards lower or higher values.

The important point is that we use a new framework, which should constitute a new line of evidence.

Comment:

23) p6, 1206: “interpretable metrics”. This is an important point, especially within the context of ‘emergent constraints’. However, I did not find it is sufficiently addressed in the previous sections (in terms of physical interpretation).

Reply:

As mentioned earlier in our response to major issues, we think the evaluation of the emergent network is an implicit evaluation of the model’s pattern effect. We do not link metrics to a specific climate feedback, nor a specific physical phenomenon, but emergent properties are a convolution of the myriad of phenomena and scales in the simulations. What matters is that the metric values reflect the model’s behaviors. *WWD* and *D_{ACE}* both contribute to assess the distance of the regional SST anomalies to the observed regional SST anomalies. Consequently, the CMIP6 models are discriminated for their SST patterns and connectivity patters, which are known to be linked to climate sensitivity estimates on long time-scales.

Comment:

24) p6, 1229: not sure “global mean” is appropriate.

Reply:

We correct in the main text:

p7, 1267-268: “The data are pre-processed with Climate Data Operator¹³ to obtain SST monthly anomalies over the time period 1975-2014.”

Comment:

25) p6, 1231: 1975-2015 corresponds to 41 years, thus 492 months. Am I missing something? This would help to talk about ‘time samples.

Reply:

The year 2015 is actually excluded from the analysis, which is conducted over the time period 1975-2014. Our time series correspond to 40 years i.e. 480 months. We correct in the main text:

p7, 1267-268: “The data are pre-processed with Climate Data Operator ¹³ to obtain SST monthly anomalies over the time period 1975-2014.”

Comment:

26) p8, 1323-324: I was not able to understand how you compute this total causal effect between two domains.

Reply:

It is an error, and “sum” should be replaced by “average” here.

Indeed, in a network of N regions, we can measure the total causal effects of each region on the $N - 1$ other regions. The Average Causal Effect is the average of the $N - 1$ total causal effects.

We correct in the main text:

p9, 1355-367: “ As mentioned earlier, we assume a linear model based on the reconstructed time series graph, and we estimate the linear causal effect (CE) of perturbations ¹⁴. These linear causal effects are normalized measure of the strength of the causal dependencies. A hypothetical propagation in a causal network may propagate along direct and indirect causal paths. There is one path coefficient for a direct causal path and several path coefficients for an indirect causal path. For a causal link $X_{t-\tau}^i \rightarrow X_t^j$, the path coefficient is defined as the standardized regression coefficient between the components X_t^j and $X_{t-\tau}^i$, and representing the expected change of X^j at time t if X^i was perturbed at time $t - \tau$ by one standard deviation. Unlike co-variances, causal effects are asymmetric measures and follow the direction of the arrows. The total causal effect between two regions i and j at a specific lag τ (measured in months) is denoted $I_{i \rightarrow j}^{CE}(\tau)$ and is computed as the sum of the product of the causal effects along each causal path between i and j ¹⁴. All causal paths between two regions are included in the measure of the total causal effect, and it is possible that causal effects along the different paths have different signs. By extension, the Average Causal effect (*ACE*) is defined as the average of the outgoing total causal effects of a given node, and indicates how important is a region as driver of the variability in the system.”

Comment:

27) p9, 1329: “randomly permuted maps”. It is unclear how you use them. Do you perform some kind of iteration, as in a bootstrap approach?

Reply:

There is no iteration in this procedure. Each grid cell is labeled either with the *ACE* value of the region it belongs to, either with a *NaN* value (if the grid cell doesn't belong to any region). The grid cells of both the spatial grid and the reference spatial grid are permuted randomly (we use the command line `numpy.random.permutation` in python). We initialized the random number generator with a specified seed for the spatial grid and another specified seed for the reference spatial grid, so that the grids have a different permutation. The seeds allow to apply the same permutations to all simulation datasets, which is important to ensure reproducibility in the code.

The numerator corresponds to the sum of the absolute error of *ACE* values, and evaluates how well the regional assignments match between the spatial grid and the true spatial grid. The denominator serves as a

way to assess the global errors by introducing randomness in the assignment of ACE values to grid cells. Finally, D_{ACE} indicates how much the regional assignments deviate from what would be expected by chance.

We modify in the text:

p9, 1369-373: “The grid cells in the map of the simulation and in the reference map are also randomly permuted, and the distance between these permuted maps (noted M_{sim} and M_{obs}) is also calculated to serve as denominator (equation (4)). Values of D_{ACE} near 0 indicate a good agreement between the ACE assigned to regions and the truth values, while high values of metric D_{ACE} indicate the ACE values assigned to the regions are close to the ones obtained by chance.”

References

1. Williamson, M. S. *et al.* Emergent constraints on climate sensitivities. *Rev. Mod. Phys.* **93**, 025004 (2021).
2. Brunner, L. *et al.* Reduced global warming from CMIP6 projections when weighting models by performance and independence. *Earth System Dynamics* **11**, 995–1012 (2020).
3. Rugenstein, M., Zelinka, M., Karnauskas, K., Ceppi, P. & Andrews, T. Patterns of Surface Warming Matter for Climate Sensitivity. *Eos* **104**, (2023).
4. Dong, Y. *et al.* Intermodel Spread in the Pattern Effect and Its Contribution to Climate Sensitivity in CMIP5 and CMIP6 Models. *Journal of Climate* **33**, 7755–7775 (2020).
5. Andrews, T., Gregory, J. M. & Webb, M. J. The Dependence of Radiative Forcing and Feedback on Evolving Patterns of Surface Temperature Change in Climate Models. *Journal of Climate* **28**, 1630–1648 (2015).
6. Gregory, J. M. *et al.* A new method for diagnosing radiative forcing and climate sensitivity. *Geophysical Research Letters* **31**, (2004).
7. Knutti, R., Rugenstein, M. A. A. & Hegerl, G. C. Beyond equilibrium climate sensitivity. *Nature Geosci* **10**, 727–736 (2017).
8. Calvin, K. *et al.* *IPCC, 2023: Climate Change 2023: Synthesis Report. Contribution of Working Groups I, II and III to the Sixth Assessment Report of the Intergovernmental Panel on Climate Change [Core Writing Team, H. Lee and J. Romero (Eds.)]. IPCC, Geneva, Switzerland.* <https://www.ipcc.ch/report/ar6/syr/> (2023) doi:10.59327/IPCC/AR6-9789291691647.
9. Mauritsen, T. *et al.* Tuning the climate of a global model. *Journal of Advances in Modeling Earth Systems* **4**, (2012).
10. Wills, R. C. J., Dong, Y., Proistosescu, C., Armour, K. C. & Battisti, D. S. Systematic Climate Model Biases in the Large-Scale Patterns of Recent Sea-Surface Temperature and Sea-Level Pressure Change. *Geophysical Research Letters* **49**, e2022GL100011 (2022).
11. Runge, J. *et al.* Inferring causation from time series in Earth system sciences. *Nat Commun* **10**, 2553 (2019).
12. Bony, S. & Dufresne, J.-L. Marine boundary layer clouds at the heart of tropical cloud feedback uncertainties in climate models. *Geophysical Research Letters* **32**, (2005).

13. Schulzweida, U., Kornblueh, L. & Quast, R. CDO user guide. *Climate data operators, Version 1*, 205–209 (2006).
14. Runge, J. *et al.* Identifying causal gateways and mediators in complex spatio-temporal systems. *Nat Commun* **6**, 8502 (2015).

REVIEWER COMMENTS

Reviewer #1 (Remarks to the Author):

Review of Revised Manuscript

netEC: an emergent network constraint on climate sensitivity. L. Ricard, F. Falasca, J. Runge, and A. Nenes.

Recommendation: Publication with major revisions.

The author responses to both reviews and their limited text revisions have clarified their objectives and analysis considerably. However, major issues remain, as detailed below, requiring further revision.

Major Issues

1. Notwithstanding the reply to reviewer 1 that the paper now refrains from the term “emergent constraint:” (a) The paper’s title includes “emergent network constraint.” (b) Ll. 7-8 state “Our analysis uses emergent networks...to constrain climate sensitivity.” (c) In Ll. 10-11 appears “derive an emergent constraint.” The term “emergent constraint” is very well established in the literature as meaning that a relationship has been demonstrated in a model between an observed behavior and a future behavior. The revised paper needs to go further in distinguishing its proposed constraint from an “emergent constraint,” as the relationship has not been demonstrated in the models analyzed.

2. Regardless of terminology, the paper asserts that the two SST networks constructed (regional time series and regional three-month time-lagged correlations) are related to the model equilibrium climate sensitivity (ECS) and transient climate response (TCR). In the Introduction, the authors provide plausible conjectures for this being so but fail to show it is true in the models they have analyzed. I appreciate the practical difficulty in their doing so, as even with their reduced dimensions the networks do not readily lend themselves to a non-arbitrary functional form or index that could be compared with model ECS/TCR. There is little doubt that SST patterns are important for ECS/TCR. But the relationship between the particular networks constructed here and ECS/TCR is not clear. Absent evidence within the model context that the SST networks are indeed related to ECS/TCR, there are significant limitations on the extent to which the agreement of the model SST networks with observations can be claimed to be a constraint.

0. The revised argument (ll. 170-173) that historical performance indicates better physics or tuning, thereby predicting future performance, has major limitations, as a couple of recent studies show. (a) Rasp *et al.* (2018, *PNAS*) constructed a computationally expensive “super-parameterized” version of an atmospheric general circulation model for present-day (PD) conditions and used it as a reference for a machine-learned (ML) version. The ML version outperformed the model using its regular parameterizations for PD. The authors also constructed a globally warmed “super-parameterized” model as a “future reference.” The ML globally warmed version, trained on PD, performed poorly relative to the model with its regular parameterizations, i.e., the PD version of model that performed best did not do so for the “future.” (If the model was retrained including the “future,” it could do well for both.) (b) Zhu *et al.* (2022, *J. Adv. Model. Earth Syst.*) show a series of generations of a coupled climate model. All but the most advanced version are able to reasonably simulate the Last Glacial Maximum. The most recently developed version, with more favorable pre-industrial to PD behavior generally, is unable to do so. The point here is that models developed using historical observations have been optimized (“tuned”) and, to some extent, structured in ways that do not guarantee “out of sample” performance consistent with historical performance. Relying on historical fidelity to observations alone, as has been done using the networks in this paper, is open to this problem.

1. Despite the major issues above, the analysis method presented here is valuable and should be published. There are indeed serious problems with the evolution of SST patterns in Earth system models, and the networks developed here provide powerful diagnostics, whether or not they can robustly constrain ECS/TCR. As my first review indicated, one publication path would be to emphasize the networks as diagnostic tools. If the authors feel strongly there is value in using agreement of the model and observed networks to weight model ECS/TCR, the limitations of doing so should be clearly stated in a revised manuscript. The endeavor might be described along the lines of “constraining climate sensitivity using SST patterns,” which would capture its physical essence and convey that it is a different approach from what are generally regarded as “emergent constraints.”

0. l. 56: Sherwood *et al.* (2020, *Rev. Geophys.*) argue that fully coupled CMIP models do not always represent “our best understanding” of

feedbacks. For two of the three largest cloud feedbacks, they assert that the representations of the physical processes in coupled climate models are inadequate to estimate feedbacks. They instead estimate these feedbacks using theory, observations, and scaled-up process-resolving models as lines of evidence. Relevant here is that SST pattern errors and errors in surface energy fluxes related to unrealistic physical representation of clouds (radiative feedbacks) in climate models are closely related problems.

Minor Point

1. ll. 91-92: My concern with the aerosol climate forcing is that (per IPCC AR6) it potentially remains sufficiently large in magnitude, even during 1975-2014, with regional patterns that would impact SST patterns, to impact the analysis that is presented here. The forcing being reduced relative to earlier time periods, which seems to the point in the text, does not eliminate this problem.

Reviewer #2 (Remarks to the Author):

I reviewed the first version of the present manuscript. The authors answered reasonably well to my questions, and I do thank them for their detailed answers and updates to the manuscript. I still have a few (minor) comments, which I think can be easily addressed (see attached document).

[Editorial Note: This document is displayed over the next three pages]

Reviewer #2 (Remarks on code availability):

This is only a readme file including the procedure to follow to reproduce the result. This does not provide much more information than what is in the manuscript. I do not think this is enough to fully reproduce the results of the manuscript.

**Title: netRC: an emergent constraint on Climate Sensitivity
based on network analysis.**

Authors: Ricard, L., F. Falasca, J. Runge, and A. Nenes

Overview

I reviewed the first version of the present manuscript. The authors answered reasonably well to my questions, and I do thank them for their detailed answers and updates to the manuscript. I still have a few (minor) comments, which I think can be easily addressed.

Comments

1. p2-3, 149-63: the authors argue that the emergent network of SSTs and the way climate models capture it are physically relevant for the ECS and TCR. They invoke the pattern effect, which occurs on rather slow time scales. Though, their emergent network only consider rather fast time scales (up to 3 months), and in the end, the metrics introduced to quantify the similarity between models and observation do not constrain so much ECS and TCR. So it remains difficult to be convinced by such statements. Of course, these statements are physically sound, but, in my opinion, they should rather be considered as working hypotheses, which can be stated more evidently as such, and which can be discussed further at the end of the manuscript, in the light of the authors' results. Note this point was part of my major comments on the first version of the manuscript. The authors did add important clarification and interesting discussion on it. I am just feeling that the wording is overly confident, and that further thought on this point would be a really nice addition in the discussion section.
2. Related to my previous comment, I am wondering to what extent the authors' emergent network based on fast time scales can render the ability of models to simulate longer timescales (e. g., ENSO, IOD, SST decadal or multi-decadal variability), which can still be observed, or tested in a perfect model approach. This is a suggestion (not for

this manuscript) that I leave to the authors' thought, and which may provide further argument that short time scales are relevant for longer ones.

3. p4, l108: the acronym *WWD* has been introduced yet.
4. p4, l109: unclear what you mean with 'self-organized deep convection'. 'Convectively-active areas' is likely sufficient here.
5. p5, l154 and elsewhere: you are using the word 'reanalysis' for reference SST dataset. I would keep to 'reference' or 'reference dataset', as 'reanalysis' is more often used in another context.
6. p6, l218: I think the 'that' can be removed.
7. p9, l334-335: Because the metrics D_{NMI} and D_{F_1} are not used at all in the present work, I do not think they deserve one page of description in the supplementary. References would be sufficient for the interested readers.
8. p9, l348: My understanding is that the first T_{ref} in the line should rather be T_{sim} .
9. There are quite a few typos across the manuscript: missing or additional spaces, two dots instead of a single one, etc. Please read carefully the entire manuscript and supplementary to remove them, even though I guess this will be addressed during the technical edition of the paper.
10. Figure 3: *WWD* and D_{ACE} are mentioned in the caption, but are not relevant for the content on the figure. I would not mention them here.
11. Figure 6: The last sentence of the caption sounds awkward. I only see the list of models, guessing they are ordered according to their rank with respect to the skill metric. Besides, what are the numbers in the parentheses after each model name?
12. Supplementary, p2, l36-38: this looks rather like an important point indicating that your weighting constraint on ECS/TCR is not much different from an unweighted approach. Could you provide an appropriate argument for relaxing this criterion?
13. Supplementary, Figure 3: I did not get how you obtained '17 measures in regions' in the first line of the figure.
14. Supplementary: Figures 10 and 11 were overlapping in the document I got.

15. Supplementary Table 1: The title in the legend does not seem to be the right one (same as Table 2)
16. Supplementary Table 2: there was two tables (lignes 216-217) and it is unclear what are their differences, or if only one should be here.
17. Supplementary Table 5: the title line (in-between lignes 243 and 244) in the table looks misplaced (or should be removed). It is also absent for Table 6.

Response to Anonymous Referee #1, NCOMMS-23-42799 reviews

We would like to thank Anonymous Referee #1 for their careful consideration of our manuscript. We sincerely appreciate your valuable comments and suggestions, which helped us again to improve the quality of the manuscript. We took consideration of their comments and remarks and modified the manuscript as detailed below.

Comment: 1. Notwithstanding the reply to reviewer 1 that the paper now refrains from the term “emergent constraint:”

(a) The paper's title includes “emergent network constraint.”

(b) Ll. 7-8 state “Our analysis uses emergent networks...to constrain climate sensitivity.”

(c) In Ll. 10-11 appears “derive an emergent constraint.”

The term “emergent constraint” is very well established in the literature as meaning that a relationship ... not been demonstrated in the models analyzed.

Reply: We have now removed any reference to the term “emergent constraint”, and replace it by the term “network-based constraint”. The title becomes “netCS: a network-based constraint to evaluate Climate Sensitivity” and the parameterization is now called “netCS”. We further discuss the differences between an Emergent Constraint (EC) and netCS, which involve the following points: *i*) An EC consists in expliciting a significant statistical relationship between an observable and the climate sensitivity, and provides a tighter estimate of climate sensitivity. The EC is therefore based on the ability of models to represent one specific variable (the “observable”), and, *ii*) a network-based constraint consists is based on the distance of their patterns and connectivity patterns to reference datasets, which provides a weighted estimate of ECS/TCR. The netCS is therefore not based on one observable, but rather on distance values which reflect different aspects of the climate models.

We include these points in the main manuscript, notably lines 64-68:

“Unlike ECs, the network-based constraint relies on multiple metrics that characterize the network. Hence, netCS does not converge necessarily towards one group of CMIP6 models (i.e. low, intermediate or high climate sensitivity group), but can discriminate models with less plausible SST and connectivity patterns, and to the extent that the latter affect ECS/TCR, help constrain climate sensitivity.”

Comment: 2. Regardless of terminology, the paper asserts that the two SST networks constructed (regional time series and regional three-month time-lagged correlations) are related to the model equilibrium climate sensitivity (ECS) and transient climate response (TCR). In the Introduction ... there are significant limitations on the extent to which the agreement of the model SST networks with observations can be claimed to be a constraint.

Reply: We do not present an emergent constraint, hence we do not focus on an explicit relationship between the SST network and climate sensitivity, but use the networks to exclude some models as less plausible. In other words, our goal is not to provide a unique likely (central estimate) value of ECS/TCR but pick those models that have the right patterns of SST and associated temporal variability, and examine their distributions of ECS/TCR. The ability to exclude models is the discriminating power of netCS, and the resulting bimodal distribution, with peaks at both low and high values of ECS/TCR we feel is an important result that contributes a new line of evidence using our new methodology.

Comment: 3. The revised argument (ll. 170-173) that historical performance indicates better physics or tuning, thereby predicting future performance, has major limitations, as a couple of recent studies show.

(a) Rasp et al. (2018, PNAS) constructed a computationally expensive “super-parameterized” version ... performed best did not do so for the “future.” (If the model was retrained including the “future,” it could do well for both.)

(b) Zhu et al. (2022, J. Adv. Model. Earth Syst.) show a series of generations ... using the networks in this paper, is open to this problem.

Reply: Thank you for these great references and important points raised! In response:

1. We do not extrapolate using netCS. We evaluate the ability of physically-based models to reproduce the historical period for which reanalysis data is available for. Therefore, netCS will tell us that the models for which the set of

physics and parameterizations are the best are the one reproducing the SST patterns. As we show, those models belong likely to the high ECS or low ECS groups.

2. If only we had reanalysis data as far back as the LGM to run netCS with! Although we do not disagree with point (b), we never claimed that we should rely solely on historical data to evaluate highly tuned models; we cannot however assess which paleoclimate models do a better job than IPCC models evaluated here, at least for projecting over the next decades (nevertheless, Falasca et al., 2022 showed that network analysis can unravel the drivers of variability at millennial timescales). Therefore, we do not want to ignore the usage of the very extensive observational datasets to exclude IPCC models that least reproduce the detrended patterns of SST (which are driven more by model physics and less by tuning).

We will add a discussion that points to the references above and bring up these important considerations 1 175-179:

“The performance of the models over the historical period is an important step towards constraining the ECS and TCR, as models that reproduce the past period could contain better physics (or are better tuned) although not guaranteed to be the best model for the future^{34,35}, provides an important line of evidence especially if evaluation is done based on climate-relevant parameters that are less subject to tuning, such as detrended patterns of SST.”

Comment:4. Despite the major issues above, the analysis ... regarded as “emergent constraints.”

Reply: Thank you for this positive assessment. We understand the concerns expressed. The truth is, this is the first application of such research; at this early stage of methodology development we already show an ability to constrain the distribution of ECS and TCR. There are many other possibilities of refining and further developing the method, that will be the topic of future work (which include suggestions already mentioned above). We hope that this discussion along with redaction of the term “emergent constraint” will provide a more balanced review of the method.

We revised the last paragraph of the discussion, lines 259-262:

“More work can focus on expanding netCS to include slower interactions, other climate-relevant property networks, multiple time periods, climate regimes or focus on different aspects and metrics of global change. We therefore encourage the climate and data science community to build upon our findings and tools to further develop network-based constraints for climate science applications.”

Comment: 5. l. 56: Sherwood et al. (2020, Rev. Geophys.) argue that fully coupled CMIP models do not always represent “our best understanding” of feedbacks. For two of the three largest cloud feedbacks, they ...unrealistic physical representation of clouds (radiative feedbacks) in climate models are closely related problems.

Reply: This is a good point, and we are in complete agreement. We focus on models with SST patterns which best reproduce the observational datasets. We revised our sentence to give a more balanced view lines 56-57: “In other words, the emergent networks and their statistics can be used to evaluate the climate system feedbacks in fully coupled CMIP models (Knutti et al., 2017)”

Comment: 1. ll. 91-92: My concern with the aerosol climate forcing is that (per IPCC AR6) it potentially remains ...the point in the text, does not eliminate this problem.

Reply:

We agree here. The aerosol forcing uncertainty in this time period can not be eliminated but it is certainly reduced, and we thought it was a good point to start from. Other periods, which are subject to higher uncertainty, will be the topic of future study.

References

Falasca, F., Crétat, J., Bracco, A., Braconnot, P., and Marti, O.: Climate change in the Indo-Pacific basin from mid- to late Holocene, *Clim Dyn*, 59, 753–766, <https://doi.org/10.1007/s00382-022-06153-z>, 2022.

Knutti, R., Rugenstein, M. A. A., and Hegerl, G. C.: Beyond equilibrium climate sensitivity, *Nature Geosci*, 10, 727–736, <https://doi.org/10.1038/ngeo3017>, 2017.

Response to Anonymous Referee #2, NCOMMS-23-42799 reviews

We would like to thank Anonymous Referee #2 for their constructive comments which contribute to improve the quality of the manuscript. Thank you for your time spent on reviewing carefully the manuscript and its supplementary information. We took consideration of their comments and remarks and modified the manuscript as detailed below.

Comment:

1. p2-3, 149-63: the authors argue that the emergent network of SSTs and the way climate models capture it are physically relevant for the ECS and TCR. They invoke the pattern effect, which occurs on rather slow time scales. Though, their emergent network only considers rather fast time scales (up to 3 months), and in the end, the metrics introduced to quantify the similarity between models and observation do not constrain so much ECS and TCR. So, it remains difficult to be convinced by such statements. Of course, these statements are physically sound, but, in my opinion, they should rather be considered as working hypotheses, which can be stated more evidently as such, and which can be discussed further at the end of the manuscript, in the light of the authors' results. Note this point was part of my major comments on the first version of the manuscript. The authors did add important clarification and interesting discussion on it. I am just feeling that the wording is overly confident, and that further thought on this point would be a really nice addition in the discussion section.

Reply:

We take note this point was already raised in your previous major comments. We now give a more balanced review of netCS. We emphasize now the primary aim of netCS is to provide a new distribution in which we eliminate models that do not reproduce well SST patterns and connectivity patterns (see revised title, and changes in the main paper). We do not provide a single likely value of ECS or TCR, and netCS power rather lies in its ability to exclude models (11 models out of 27 models are excluded for the estimation of ECS). Still excluding models is very valuable because when combined with other lines of evidence, it provides the true constraint on ECS/TCR.

About the fast time scales:

It is a very interesting comment. The importance of the two-way interactions between the tropical and midlatitudes regions on intraseasonal time scales of 10 to 100 days is well recognized (Stan et al., 2017). We agree we only focused on the SST perturbations that propagate at fast time scales (up to 3 months), which means we assumed the representation of fast teleconnections is necessary for the representation of longer teleconnections, and in that regard the networks are linked to the (slow) pattern effect. Fast teleconnections have “echoes” that are also relevant to the potential propagations at longer timescale. We also recall these fast teleconnections are evaluated over a time period of forty years, which means we also evaluate their persistence over time, and we showed these fast teleconnections (through D_{ACE}) contained some discriminative power.

On a practical point of view, we made the choice to work with sequences of τ , to compare the propagation over a range of time lags. We think it is relevant in a context of model evaluation, but we agree other τ_{max} could be used, at the condition we remove the “echoes” of the shorter timescales on longer timescales that are multiple of them. We worked with a small range of τ to ensure the robustness of our results. Another choice would consist to select for each link a specific τ (i.e. $\tau_{min} = \tau_{max}$), so that the causal effect value is maximized. This is the subject of an ongoing study, which focuses more on process understanding.

We discuss more these points in the main paper.

Comment:

2. Related to my previous comment, I am wondering to what extent the authors' emergent network based on fast time scales can render the ability of models to simulate longer timescales (e. g., ENSO, IOD, SST decadal or multi-decadal variability), which can still be observed, or tested in a perfect model approach. This is a suggestion (not for this manuscript) that I leave to the authors' thought, and which may provide further argument that short time scales are relevant for longer ones.

Reply:

This is a great comment and we are currently working on this. As implied in our response to the comment above there is an imprint of the fast timescales on the longer responses. To unravel this requires work and this is the topic of a future study. These processes are seen on other components of the climate system e.g. clouds are really short time-lived and aerosol-cloud interactions are short timescales but their imprint on climate and climate variability is evident.

3. p4, l108: the acronym WWD has been introduced yet.

Thanks, it is corrected.

4. p4, l109: unclear what you mean with 'self-organized deep convection'. 'Convectively-active areas' is likely sufficient here.

Thanks for the suggestion, we made the change.

5. p5, l154 and elsewhere: you are using the word 'reanalysis' for reference SST dataset. I would keep to 'reference' or 'reference dataset', as 'reanalysis' is more often used in another context.

We replaced reanalysis dataset by reference dataset (excepted the first occurrence, for which we introduce the term "reference dataset")

6. p6, l218: I think the 'that' can be removed.

Thanks, it is removed.

7. p9, l334-335: Because the metrics D_{NMI} and D_{F_1} are not used at all in the present work, I do not think they deserve one page of description in the supplementary. References would be sufficient for the interested readers.

We can understand that comment but we prefer to incorporate them for completeness for the reader.

8. p9, l348: My understanding is that the first T_{ref} in the line should rather be T_{sim}

Indeed, it's an error. We now replace T_{obs} and M_{obs} by T_{ref} and M_{ref} for more clarity.

9. There are quite a few typos across the manuscript: missing or additional spaces, two dots instead of a single one, etc. Please read carefully the entire manuscript and supplementary to remove them, even though I guess this will be addressed during the technical edition of the paper.

Thanks. We tried to remove as much as possible.

10. Figure 3: WWD and D_{ACE} are mentioned in the caption, but are not relevant for the content on the figure. I would not mention them here.

We agree it is better without the mention of WWD and D_{ACE} .

11. Figure 6: The last sentence of the caption sounds awkward. I only see the list of models, guessing they are ordered according to their rank with respect to the skill metric. Besides, what are the numbers in the parentheses after each model name?

The numbers in the parentheses are the number of runs per model. We change the last sentence:

“The legend shows the list of models sorted by decreasing order of weights i.e. from most important one (1) to less important one (27). The numbers in the parentheses indicate the number of runs per model.”

12. Supplementary, p2, 136-38: this looks rather like an important point indicating that your weighting constraint on ECS/TCR is not much different from an unweighted approach. Could you provide an appropriate argument for relaxing this criterion?

This is a good comment. With a threshold of 80% we did not obtain very different weights from the equal weights. We agree that the selection of this threshold could be subject to debate in case we present an Emergent Constraint, because the constrained range is linked to the choice of the threshold. However, we know our two metrics do not favor one particular group of models (among low, intermediate, high climate sensitivity models), which limits the convergence of the range towards one central value, and we rather aimed to emphasize these clusters of models. To this end, we decided to relax the criterion to obtain an interesting gradient of weights, which allows to discriminate the models (Supplementary Figure 6). Besides, a threshold of 70% remains high, and still shows netCS has a discriminating power.

13. Supplementary, Figure 3: I did not get how you obtained ‘17 measures in regions’ in the first line of the figure.

We apologize for the confusion. The number should indicate the number of regions. As this number is different for the two metrics, we now prefer to simply indicate “number of regions”.

14. Supplementary: Figures 10 and 11 were overlapping in the document I got.

Ok, we make sure it is not the case now.

15. Supplementary Table 1: The title in the legend does not seem to be the right one (same as Table 2)

Thanks, the title is now: “Network metric values of CMIP6 models”.

16. Supplementary Table 2: there were two tables (lines 216-217) and it is unclear what are their differences, or if only one should be here.

One table was from the first version of the paper, and the other one from the revised version. Somehow the old table was not removed. We removed it.

17. Supplementary Table 5: the title line (in-between lines 243 and 244) in the table looks misplaced (or should be removed). It is also absent for Table 6.

I removed the title, as we indicate in the legend whether it is ECS estimates (Supplementary Table 5) or TCR estimates (Supplementary Table 6).

References

Stan, C., Straus, D. M., Frederiksen, J. S., Lin, H., Maloney, E. D., and Schumacher, C.: Review of Tropical-Extratropical Teleconnections on Intraseasonal Time Scales, *Reviews of Geophysics*, 55, 902–937, <https://doi.org/10.1002/2016RG000538>, 2017.

REVIEWERS' COMMENTS

Reviewer #1 (Remarks to the Author):

I recommend publication of the most recent version. The utility of netCS as an analysis tool is conveyed well, and both the insights from, and limits of, its use in narrowing estimates of climate sensitivity (CS) are expressed more clearly than in earlier versions. Publication of the network analysis method could benefit an important recent discussion in the climate community in the wake of Wills (2022, Geophys. Res. Lett.), which shows climate model large ensembles simulate SST pattern trends from 1979 to 2020 quite poorly. The Wills focus is on longer-time pattern trends, clearly relevant for global warming, with pattern analysis methods differing greatly from those used here. I would leave to the authors whether they wish to comment on these recent developments in this paper. It is quite possible the greatest contribution from this paper will be the "net" method applied to a wide range of climate studies, as opposed to the possible "CS" constraint.

An edit for minor changes to enhance clarity is suggested. For example, ll. 55-56, the meaning of the phrase "in warming patterns" is not clear.